# CNSP: Consistent Null-Space Projection for Principled Prompt-Based Continual Learning

## Abstract

Continual learning aims to acquire new knowledge sequentially without forgetting previous tasks, yet catastrophic forgetting remains a major challenge. Prompt-based continual learning has recently shown competitive empirical progress, yet its theoretical underpinnings remain incomplete. We introduce Consistent Null-Space Projection (CNSP), the first unified and mathematically rigorous framework for representational consistency in prompt-based continual learning. CNSP proves that task-performance preservation reduces to two jointly sufficient requirements—feature preservation and head preservation—while deriving explicit consistency conditions under full Transformer parameterization. These conditions yield a tractable null-space projection rule for stable prompt updates. Across various benchmarks and backbones, CNSP demonstrates consistent improvements in accuracy and forgetting, with especially clear benefits in high-dimensional and domain-shift scenarios.

## 1 Introduction

Continual learning seeks to enable models to acquire new knowledge from sequential tasks without forgetting previously learned knowledge. Despite significant progress, catastrophic forgetting (Mc-Closkey & Cohen, 1989), where performance on old tasks deteriorates when learning new ones, remains a major challenge (Wang et al., 2024).

Recently, prompt-tuning methods, particularly Visual Prompt Tuning (VPT) (Jia et al., 2022) with Vision Transformers (ViTs) (Dosovitskiy et al., 2020), have emerged as a promising rehearsal-free solution for continual learning (Wang et al., 2024). By updating only lightweight prompts while freezing the backbone, these methods achieve both computational and storage efficiency. While several prompt-based methods for continual learning (e.g., L2P (Wang et al., 2022b), DualPrompt (Wang et al., 2022a), Coda-Prompt (Smith et al., 2023)) have demonstrated competitive empirical performance, theoretical foundations of prompt-based continual learning remain underdeveloped. Classical orthogonal-projection methods (Saha et al., 2021; Deng et al., 2021; Wang et al., 2021) guarantee retention by deriving explicit consistency conditions in linear or convolutional networks, but these guarantees do not transfer to Transformers: nonlinear LayerNorm, learned QKV projections, and multi-head attention invalidate linear assumptions.

To bridge this theoretical gap, recent efforts have adapted projection ideas to ViTs/VPT (Qiao et al., 2024; Lu et al., 2024). PGP (Qiao et al., 2024) approximates Multi-Head Self-Attention (MHSA) and LayerNorm as linear mappings and constructs orthogonal subspaces in the sum space of inputs and prompts, but its strong simplification limits generality. $NSP^2$ (Lu et al., 2024) advances this line of work by deriving approximate sufficient conditions for preserving representations under prompt updates. However, $NSP^2$ still leaves several fundamental aspects of the Transformer pipeline theoretically unresolved: the extension from single-head to multi-head attention lacks principled justification; LayerNorm's broadcasted affine structure is simplified in ways that obscure its algebraic properties; strong assumptions on prompt distribution (mean–variance invariance) can introduce instability; and the classification head is omitted from the overall consistency formulation. Consequently, existing analyses offer valuable local insights but do not yield an end-to-end account of how prompt updates interact with the entire Transformer computation to preserve task performance.

Motivated by the absence of a complete theoretical closure, we revisit prompt-based continual learning from an end-to-end, matrix-level perspective and introduce Consistent Null-Space Projection (CNSP)—a principled framework that establishes the first unified representational-consistency formulation for prompted Transformers in continual learning. Our central insight is that performance preservation admits a clean mathematical decomposition into two jointly sufficient components: *feature preservation*, ensuring that MHSA+LayerNorm representations for previous tasks remain stable, and head preservation, guaranteeing decision-level consistency for previously learned classifier(s). Building on this decomposition, we derive sufficient consistency conditions for preserving past-task performance by jointly accounting for the coupled roles of MHSA, LayerNorm, and classification heads under full prompted Transformer parameterization, from which a tractable null-space projection rule emerges as an analytical consequence.

Our contributions are threefold:

(i) **Unified theoretical closure.** We establish the first unified and mathematically rigorous framework demonstrating that prompt-based continual learning reduces to a single consistency principle integrating feature preservation and head preservation.

(ii) **Sufficient consistency conditions.** By rigorously modeling MHSA, LayerNorm, and classifier behavior using an algebraically precise matrix-form characterization under full prompted Transformer parameterization, we derive explicit sufficient conditions for past-task performance preservation, leading to a tractable null-space projection rule.

(iii) **Consistent empirical gains.** CNSP reliably reduces forgetting and improves accuracy across diverse benchmarks and backbones, with more noticeable gains in high-dimensional and cross-domain settings—all without introducing architectural heuristics or task-specific prompt designs.

## 2 RELATED WORKS

### 2.1 PROMPT-BASED METHODS FOR CONTINUAL LEARNING

Prompt-based methods have emerged as a promising paradigm for continual learning, offering lightweight task adaptation with reduced interference. Early approaches include L2P (Wang et al., 2022b), which employs a learnable prompt pool with dynamic prompt selection, and DualPrompt (Wang et al., 2022a), which further disentangles prompts into general and expert subsets to balance knowledge sharing and task-specific adaptation. CPrompt (Gao et al., 2024) addresses train–test inconsistency via classifier- and prompt-consistent learning, while CODA-Prompt (Smith et al., 2023) decomposes prompts into fine-grained components and dynamically recombining them through attention. EvoPrompt (Kurniawan et al., 2024) formulates prompt learning as evolutionary search for long-term adaptability. Beyond purely visual prompts, LGCL (Khan et al., 2023) incorporates semantic guidance by aligning visual representations with language embeddings at both task and class levels. ConvPrompt (Roy et al., 2024) leverages convolutional structures to construct hierarchical prompts and employs large language models to estimate task similarity.

More recent efforts pursue principled formulations. PGP (Qiao et al., 2024) enforces gradient orthogonality in the joint input–prompt space by linearizing the nonlinear effects of MHSA and LayerNorm, while NSP$^2$ (Lu et al., 2024) derives null-space projection conditions for VPT by explicitly analyzing attention and LayerNorm. CPG (Lu et al., 2025) extends this direction with consistency-constrained mixtures of experts.

Prompt-based continual learning methods are versatile and effective, but their theoretical foundations remain limited. NSP$^2$ advances this line via null-space projection for VPT, yet its simplified treatment of multi-head attention and LayerNorm reduces robustness and interpretability. We address these gaps by formalizing MHSA and LayerNorm in exact matrix form and deriving sufficient conditions for performance preservation, thus yielding a principled and effective framework.

### 2.2 ORTHOGONAL PROJECTION METHODS FOR CONTINUAL LEARNING

Orthogonal projection methods have been widely studied in CNNs and MLPs (e.g., OWM (Zeng et al., 2019), GPM (Saha et al., 2021), Adam-NSCL (Wang et al., 2021)) as a principled way to reduce task interference. The core idea is to preserve past feature responses by constrain-

ing parameter updates within subspaces orthogonal to previous tasks' features, formalized as $\boldsymbol{X}^{\top}(\boldsymbol{\Theta} + \Delta\boldsymbol{\Theta}) = \boldsymbol{X}^{\top}\boldsymbol{\Theta}$, where $\boldsymbol{X}$ denotes input features, $\boldsymbol{\Theta}$ the convolutional or linear parameters, and $\Delta\boldsymbol{\Theta}$ the update. Enforcing $\boldsymbol{X}^{\top}\Delta\boldsymbol{\Theta} = \boldsymbol{0}$ guarantees invariance of past intermediate representations, thereby retaining knowledge. This principle, rooted in gradient limitation theory, provides a mathematical explanation of the stability-plasticity trade-off. Representative methods include GPM Saha et al. (2021), which projects gradients onto the null space of previously learned inputs; TRGP (Lin et al., 2022b), which refines GPM by scaling task gradients with a trust region matrix before projection; and Connector (Lin et al., 2022a) interpolates a normally updated model with one constrained by projection.

These methods work well in linear or convolutional architectures, where orthogonality condition holds exactly. However, in Transformers, nonlinear MHSA, softmax, and LayerNorm interactions violate this linearity, making classical projections unreliable. Recent works have applied orthogonal projection to prompt tuning (Qiao et al., 2024; Lu et al., 2024). NSP$^2$ (Lu et al., 2024) unfolds VPT forward propagation and derives approximate sufficient conditions for performance preservation via null-space projection. Building on this foundation, we introduce stricter and more general consistency conditions alongside stable optimization strategies, thereby strengthening projection-based continual learning in VPT.

## 3 PRELIMINARIES

In this section, we introduce the notations and conventions, describe the forward propagation process of transformer blocks under VPT-deep (Jia et al., 2022)—where learnable prompts are inserted into the input token sequence at each block for efficient adaptation with a frozen backbone—and formalize the continual learning problem in this setting.

### 3.1 NOTATIONS

We use the following notational conventions: (i) Non-bold letters denote positive integers, e.g., $a, A \in \mathbb{N}^{+}$. (ii) Bold lowercase denote vectors, $\boldsymbol{a} \in \mathbb{R}^n$; bold uppercase denote matrices, $\boldsymbol{A} \in \mathbb{R}^{n \times m}$. (iii) Column concatenation: for $\boldsymbol{A}^{(i)} \in \mathbb{R}^{n \times d}$, $[\boldsymbol{A}^{(i)}]_{i=1}^{N} = \begin{bmatrix} \boldsymbol{A}^{(1)} & \dots & \boldsymbol{A}^{(N)} \end{bmatrix} \in \mathbb{R}^{n \times Nd}$. (iv) Row concatenation: for $\boldsymbol{A}^{(i)} \in \mathbb{R}^{n \times d}$, $\begin{bmatrix} \boldsymbol{A}^{(1)}; \dots; \boldsymbol{A}^{(N)} \end{bmatrix} \in \mathbb{R}^{Nn \times d}$. (v) Slicing: if $\boldsymbol{A} = \begin{bmatrix} \boldsymbol{A}^{(1)}; \boldsymbol{A}^{(2)} \end{bmatrix}$ with $\boldsymbol{A}^{(1)} \in \mathbb{R}^{n \times d}$, then $\boldsymbol{A}[:n] = \boldsymbol{A}^{(1)}$.

### 3.2 FORWARD PROPAGATION IN VPT-DEEP

Let $\boldsymbol{X} \in \mathbb{R}^{(N+1) \times D}$ denote the patch embeddings of an input image, where $N$ is the number of patches, the first row is the pre-trained [CLS] token, and $D$ the embedding dimension. A standard ViT with depth $L$ can be written as $f_{\text{ViT}}(\boldsymbol{X} \mid \boldsymbol{\Theta}; \{\boldsymbol{W}, \boldsymbol{b}\})$, where $\boldsymbol{\Theta}$ are frozen backbone parameters and $\{\boldsymbol{W}, \boldsymbol{b}\}$ are learnable classification head parameters. Let $\boldsymbol{X}^{(l)} \in \mathbb{R}^{(N+1) \times D}$ denote the input embeddings of the $l$-th block.

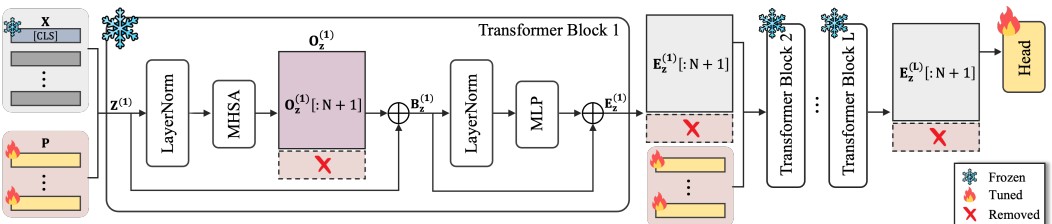

Figure 1: Forward propagation in VPT. Image embeddings ($\boldsymbol{X}$) and learnable prompt embeddings ($\boldsymbol{P}$) are jointly processed by LayerNorm, MHSA, and MLP with residual connections in each transformer block. Modules marked with ❄ are frozen, those with 🔥 are trainable, and those with ✗ indicate information discarded before entering the next layer.

In VPT-deep, each block $l$ introduces $M$ learnable prompts $\boldsymbol{P}^{(l)} \in \mathbb{R}^{M \times D}$, concatenated with image embeddings to form the input sequences: $\boldsymbol{Z}^{(l)} = \left[\boldsymbol{X}^{(i)}; \boldsymbol{P}^{(l)}\right] \in \mathbb{R}^{(N+1+M) \times D}$, $\boldsymbol{X}^{(1)} = \boldsymbol{X}$. The forward propagation in VPT-deep is illustrated in fig. 1. Each transformer block applies LayerNorm (LN), Multi-Head Self-Attention (MHSA), and an MLP with residual connections:

$$\boldsymbol{O}_{\boldsymbol{Z}}^{(l)} = \text{MHSA}(\text{LN}(\boldsymbol{Z}^{(l)})), \quad \boldsymbol{B}_{\boldsymbol{Z}}^{(l)} = \boldsymbol{Z}^{(l)} + \boldsymbol{O}_{\boldsymbol{Z}}^{(l)}, \quad \boldsymbol{E}_{\boldsymbol{Z}}^{(l)} = \boldsymbol{B}_{\boldsymbol{Z}}^{(l)} + \text{MLP}(\text{LN}(\boldsymbol{B}_{\boldsymbol{Z}}^{(l)})). \quad (1)$$

Before the next block, the prompts are removed and replaced with a new set of learnable prompts:

$$\boldsymbol{X}^{(l+1)} = \boldsymbol{E}_{\boldsymbol{Z}}^{(l)}[: N + 1], \quad \boldsymbol{Z}^{(l+1)} = [\boldsymbol{X}^{(l+1)}; \boldsymbol{P}^{(l+1)}]. \quad (2)$$

Thus, prompts modulate intermediate features at each block but do not persist across blocks. The final [CLS] token is passed into the classification head:

$$f_{\text{ViT}}\left(\boldsymbol{X} \mid \boldsymbol{\Theta}; \{\boldsymbol{W}, \boldsymbol{b}\}; \{\boldsymbol{P}^{(l)}\}_{l=1}^{L}\right) = softmax\left(\boldsymbol{E}_{\boldsymbol{Z}}^{(L)}[: 1]\boldsymbol{W} + \boldsymbol{b}\right), \quad (3)$$

where $\boldsymbol{W} \in \mathbb{R}^{D \times C}$, $\boldsymbol{b} \in \mathbb{R}^{C}$, and $C$ is the number of classes.

### 3.3 PROBLEM FORMULATION: CONTINUAL LEARNING WITH VPT-DEEP

We consider a continual learning setting with a sequence of tasks $\mathcal{T}_1, \ldots, \mathcal{T}_T$. Each task $\mathcal{T}_t$ is associated with a dataset $\mathcal{D}_t = \{(\boldsymbol{x}_t^{(i)}, y_t^{(i)})\}_{i=1}^{N_t}$, where $\boldsymbol{x}_t^{(i)}$ is an image and $y_t^{(i)}$ its label. At block $l$, let $\boldsymbol{P}_t^{(l)}$ be the prompts after learning tasks up to $\mathcal{T}_t$. For the next task $\mathcal{T}_{t+1}$, updates are:

$$\boldsymbol{P}_{t+1}^{(l)} = \boldsymbol{P}_t^{(l)} + \Delta\boldsymbol{P}^{(l)}, \quad \boldsymbol{W}_{t+1} = \boldsymbol{W}_t + \Delta\boldsymbol{W}, \quad \boldsymbol{b}_{t+1} = \boldsymbol{b}_t + \Delta\boldsymbol{b}, \quad (4)$$

The continual learning objective requires preserving performance on previous tasks after training on a new one. We formalize the objective as:

$$f_{\text{ViT}}(\boldsymbol{X}_t \mid \boldsymbol{\Theta}; \{\boldsymbol{W}_{t+1}, \boldsymbol{b}_{t+1}\}; \{\boldsymbol{P}_{t+1}^{(l)}\}_{l=1}^{L}) = f_{\text{ViT}}(\boldsymbol{X}_t \mid \boldsymbol{\Theta}; \{\boldsymbol{W}_t, \boldsymbol{b}_t\}; \{\boldsymbol{P}_t^{(l)}\}_{l=1}^{L}), \quad (5)$$

which enforces that model outputs on old inputs $\boldsymbol{X}_t$ from $\mathcal{T}_t$ must remain unchanged. Equivalently, updates $\{\Delta\boldsymbol{P}^{(l)}\}_{l=1}^{L}$ and $\{\Delta\boldsymbol{W}, \Delta\boldsymbol{b}\}$ must not affect the predictions for previously learned tasks. To characterize this objective, we introduce two key propositions that decompose it into two complementary requirements: **feature preservation** at intermediate blocks and **head preservation** at the classification head.

**Proposition 1.** *Consider two tasks $\mathcal{T}_t$ and $\mathcal{T}_{t+1}$. If the image-token part of attention outputs remains unchanged for all blocks in VPT, i.e.,*

$$\boldsymbol{O}_{\boldsymbol{Z}_{t,t+1}}^{(l)}[: N + 1] = \boldsymbol{O}_{\boldsymbol{Z}_{t,t}}^{(l)}[: N + 1], \quad l = 1, 2, \ldots, L, \quad (6)$$

*then the image-token output of the final block is preserved:*

$$\boldsymbol{E}_{\boldsymbol{Z}_{t,t+1}}^{(L)}[: N + 1] = \boldsymbol{E}_{\boldsymbol{Z}_{t,t}}^{(L)}[: N + 1], \quad (7)$$

*which we refer to as **feature preservation**. Here $\boldsymbol{O}_{\boldsymbol{Z}_{t,t+1}}^{(l)}$, $\boldsymbol{O}_{\boldsymbol{Z}_{t,t}}^{(l)}$, $\boldsymbol{E}_{\boldsymbol{Z}_{t,t+1}}^{(L)}$, and $\boldsymbol{E}_{\boldsymbol{Z}_{t,t}}^{(L)}$ are computed by eq. (1). In particular, the final [CLS] token is preserved:*

$$\boldsymbol{E}_{\boldsymbol{Z}_{t,t+1}}^{(L)}[: 1] = \boldsymbol{E}_{\boldsymbol{Z}_{t,t}}^{(L)}[: 1]. \quad (8)$$

*Proof Sketch.* By eq. (2), since prompts are replaced at each block, their effect propagates only through the image-token outputs $\boldsymbol{E}_{\boldsymbol{Z}}^{(l)}[: N + 1]$. Row-wise independence of post-MHSA operations ensures invariance of $\boldsymbol{O}_{\boldsymbol{Z}}^{(l)}[: N + 1]$ carries forward. Full proof is provided in Appendix A.

**Proposition 2.** *Consider tasks $\mathcal{T}_t$ and $\mathcal{T}_{t+1}$. If the final [CLS] token remains unchanged (i.e., eq. (8) holds) and the head updates satisfy:*

$$\boldsymbol{E}_{\boldsymbol{Z}_{t,t}}^{(L)}[: 1]\Delta\boldsymbol{W} + \Delta\boldsymbol{b} = \boldsymbol{0}, \quad (9)$$

*which we refer to as **head preservation**, then the continual learning objective eq. (5) is satisfied.*

*Proof.* From eq. (8) and eq. (9), we have

$$\boldsymbol{E}_{\boldsymbol{Z}_{t,t+1}}^{(L)}[:1]\boldsymbol{W}_{t+1} + \boldsymbol{b}_{t+1} = \boldsymbol{E}_{\boldsymbol{Z}_{t,t+1}}^{(L)}[:1]\left(\boldsymbol{W}_t + \Delta\boldsymbol{W}\right) + \left(\boldsymbol{b}_t + \Delta\boldsymbol{b}\right) = \boldsymbol{E}_{\boldsymbol{Z}_{t,t}}^{(L)}[:1]\boldsymbol{W}_t + \boldsymbol{b}_t, \quad (10)$$

which, by eq. (3), implies eq. (5). $\qquad\square$

Together, Propositions 1 and 2 reduce continual learning preservation to two conditions: feature preservation, requiring prompt updates to leave image-token representations invariant for representational stability, and head preservation, requiring the classifier to maintain consistent mappings of the preserved [CLS] token. This decomposition of objective eq. (5) forms the theoretical basis for the sufficient conditions derived in the next section. Particularly, proposition 1 implies intermediate feature preservation reduces to ensuring invariance at the MHSA output. Therefore, in the derivations presented in the next section, we focus on LayerNorm and MHSA, as they are the only operations directly relevant to feature preservation.

## 4 METHOD

Our method builds on Propositions 1 and 2, enforcing both feature and head preservation. Feature preservation is ensured through explicit constraints on prompt updates and head preservation guaranteed by design. According to proposition 1, the preservation objective can be analyzed blockwise by focusing solely on the MHSA and its preceding LayerNorm operation. Other components (e.g., MLP) are row-wise and do not alter the preservation property. Since eq. (6) must be satisfied at every layer, we omit the superscript $(l)$) when deriving algebraic sufficient conditions on prompt updates.

### 4.1 UNFOLDING FORWARD PROPAGATION OF LAYERNORM AND MHSA IN A SINGLE BLOCK

Let the input sequence be $\boldsymbol{Z}_{t,t+1} = [\boldsymbol{X}_t; \boldsymbol{P}_{t+1}] \in \mathbb{R}^{(N+1+M)\times D}$, where $\boldsymbol{P}_{t+1} = \boldsymbol{P}_t + \Delta\boldsymbol{P}$. Figure 2 illustrates the unrolled forward pass of LayerNorm (LN) and MHSA in a single transformer block. In this setting, computations involving $\boldsymbol{Q}_{\boldsymbol{P}_{t+1}}^{(h)}$ can be safely omitted; justification is provided below.

First, $\boldsymbol{Z}_{t,t+1}$ is passed through LayerNorm, we introduce the following lemma:

**Lemma 1.** *Consider the computation of LayerNorm* $\mathrm{LN}(\cdot)$. *Let* $\boldsymbol{A} = [\boldsymbol{A}^{(1)}; \boldsymbol{A}^{(2)}]$ *with* $\boldsymbol{A}^{(1)} \in \mathbb{R}^{n\times d}$ *and* $\boldsymbol{A}^{(2)} \in \mathbb{R}^{m\times d}$. *Then* $\mathrm{LN}(\boldsymbol{A}) = [\mathrm{LN}(\boldsymbol{A}^{(1)}); \mathrm{LN}(\boldsymbol{A}^{(2)})]$.

Proof of Lemma 1 is in Appendix A. According to lemma 1, we have $\mathrm{LN}(\boldsymbol{Z}_{t,t+1}) = [\mathrm{LN}(\boldsymbol{X}_t); \mathrm{LN}(\boldsymbol{P}_t + \Delta\boldsymbol{P})]$. Next, the normalized sequence undergoes QKV transformation, including QKV projections and multi-head splitting. For head $h \in \{1, \ldots, H\}$, we have:

$$\boldsymbol{\Phi}_{\boldsymbol{Z}_{t,t+1}}^{(h)} = \mathrm{LN}(\boldsymbol{Z}_{t,t+1})\,\boldsymbol{W}_\phi^{(h)} + \boldsymbol{1}_{N+1+M}\boldsymbol{b}_\phi^{(h)\top} = \begin{bmatrix} \mathrm{LN}(\boldsymbol{X}_t)\,\boldsymbol{W}_\phi^{(h)} + \boldsymbol{1}_{N+1}\boldsymbol{b}_\phi^{(h)\top} \\ \mathrm{LN}(\boldsymbol{P}_t + \Delta\boldsymbol{P})\,\boldsymbol{W}_\phi^{(h)} + \boldsymbol{1}_M\boldsymbol{b}_\phi^{(h)\top} \end{bmatrix}, \quad (11)$$

where $\boldsymbol{\Phi} \in \{\boldsymbol{Q}, \boldsymbol{K}, \boldsymbol{V}\}$, $\phi$ serves as the symbolic index for $q, k, v$, and $\boldsymbol{W}_\phi^{(h)} \in \mathbb{R}^{D\times D_H}$, $\boldsymbol{b}_\phi^{(h)} \in \mathbb{R}^{D_H}$, and $D_H = D/H$. Let

$$\boldsymbol{Q}_{\boldsymbol{Z}_{t,t+1}}^{(h)} = \left[\boldsymbol{Q}_{\boldsymbol{X}_t}^{(h)}; \boldsymbol{Q}_{\boldsymbol{P}_{t+1}}^{(h)}\right], \quad \boldsymbol{K}_{\boldsymbol{Z}_{t,t+1}}^{(h)} = \left[\boldsymbol{K}_{\boldsymbol{X}_t}^{(h)}; \boldsymbol{K}_{\boldsymbol{P}_{t+1}}^{(h)}\right], \quad \boldsymbol{V}_{\boldsymbol{Z}_{t,t+1}}^{(h)} = \left[\boldsymbol{V}_{\boldsymbol{X}_t}^{(h)}; \boldsymbol{V}_{\boldsymbol{P}_{t+1}}^{(h)}\right], \quad (12)$$

where $\boldsymbol{Q}_{\boldsymbol{X}_t}^{(h)}, \boldsymbol{K}_{\boldsymbol{X}_t}^{(h)}, \boldsymbol{V}_{\boldsymbol{X}_t}^{(h)} \in \mathbb{R}^{(N+1)\times D_H}$ and $\boldsymbol{Q}_{\boldsymbol{P}_{t+1}}^{(h)}, \boldsymbol{K}_{\boldsymbol{P}_{t+1}}^{(h)}, \boldsymbol{V}_{\boldsymbol{P}_{t+1}}^{(h)} \in \mathbb{R}^{M\times D_H}$.

For each head $h$, the attention weights and value aggregation are computed as

$$\boldsymbol{H}_{\boldsymbol{Z}_{t,t+1}}^{(h)} = softmax\left(\boldsymbol{Q}_{\boldsymbol{Z}_{t,t+1}}^{(h)}\boldsymbol{K}_{\boldsymbol{Z}_{t,t+1}}^{(h)\top}/\sqrt{D_H}\right)\boldsymbol{V}_{\boldsymbol{Z}_{t,t+1}}^{(h)}, \quad (13)$$

where the softmax function acts on the rows of matrix $\boldsymbol{Q}_{\boldsymbol{Z}_{t,t+1}}^{(h)}\boldsymbol{K}_{\boldsymbol{Z}_{t,t+1}}^{(h)\top}$. Finally, concatenating across heads and applying the output projection gives

$$\boldsymbol{O}_{\boldsymbol{Z}_{t,t+1}} = \left[\boldsymbol{H}_{\boldsymbol{Z}_{t,t+1}}^{(h)}\right]_{h=1}^{H}\boldsymbol{W}_o + \boldsymbol{1}_{N+1+M}\boldsymbol{b}_o^\top. \quad (14)$$

By proposition 1, feature preservation requires $\boldsymbol{O}_{\boldsymbol{Z}_{t,t+1}}[: N+1] = \boldsymbol{O}_{\boldsymbol{Z}_{t,t}}[: N+1]$. Note that

$$\left[\boldsymbol{H}_{\boldsymbol{Z}_{t,t+1}}^{(h)}\right]_{h=1}^{H}[: N+1] = \left[softmax(\boldsymbol{Q}_{\boldsymbol{X}_t}^{(h)}\boldsymbol{K}_{\boldsymbol{Z}_{t,t+1}}^{(h)}{}^{\top}/\sqrt{D_H})\boldsymbol{V}_{\boldsymbol{Z}_{t,t+1}}^{(h)}\right]_{h=1}^{H}. \tag{15}$$

Therefore, only $\boldsymbol{Q}_{\boldsymbol{X}_t}^{(h)}$ contributes to the image-token outputs, terms from $\boldsymbol{Q}_{\boldsymbol{P}_{t+1}}^{(h)}$ can be omitted in the preservation analysis.

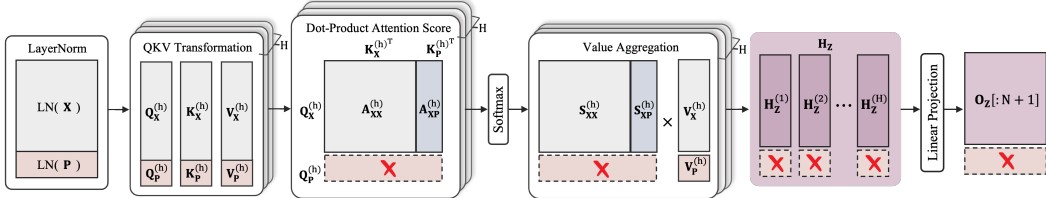

Figure 2: Detailed illustration of LayerNorm and the multi-head self-attention mechanism in a VPT transformer block, showing how image and prompt embeddings are jointly processed. Those with ✗ indicate information discarded before entering the next layer.

## 4.2 SUFFICIENT CONDITIONS FOR FEATURE PRESERVATION IN VPT

Building on the above expansion, we restrict attention to the $\boldsymbol{Q}_{\boldsymbol{X}_t}^{(h)}$-dependent pathways. Define

$$\boldsymbol{A}_{\boldsymbol{Z}_{t,t+1}}^{(h)} = \frac{\boldsymbol{Q}_{\boldsymbol{X}_t}^{(h)}\boldsymbol{K}_{\boldsymbol{Z}_{t,t+1}}^{(h)}{}^{\top}}{\sqrt{D_H}} = \frac{1}{\sqrt{D_H}}\boldsymbol{Q}_{\boldsymbol{X}_t}^{(h)}\left[\boldsymbol{K}_{\boldsymbol{X}_t}^{(h)}{}^{\top} \quad \boldsymbol{K}_{\boldsymbol{P}_{t+1}}^{(h)}{}^{\top}\right] \triangleq \left[\boldsymbol{A}_{\boldsymbol{X}_t\boldsymbol{X}_t}^{(h)} \quad \boldsymbol{A}_{\boldsymbol{X}_t\boldsymbol{P}_{t+1}}^{(h)}\right], \tag{16}$$

$$\boldsymbol{S}_{\boldsymbol{Z}_{t,t+1}}^{(h)} = softmax\left(\left[\boldsymbol{A}_{\boldsymbol{X}_t\boldsymbol{X}_t}^{(h)} \quad \boldsymbol{A}_{\boldsymbol{X}_t\boldsymbol{P}_{t+1}}^{(h)}\right]\right) \triangleq \left[\boldsymbol{S}_{\boldsymbol{X}_t\boldsymbol{X}_t}^{(h)} \quad \boldsymbol{S}_{\boldsymbol{X}_t\boldsymbol{P}_{t+1}}^{(h)}\right], \tag{17}$$

where $\boldsymbol{A}_{\boldsymbol{X}_t\boldsymbol{X}_t}^{(h)}, \boldsymbol{S}_{\boldsymbol{X}_t\boldsymbol{X}_t}^{(h)} \in \mathbb{R}^{(N+1)\times(N+1)}$ and $\boldsymbol{A}_{\boldsymbol{X}_t\boldsymbol{P}_{t+1}}^{(h)}, \boldsymbol{S}_{\boldsymbol{X}_t\boldsymbol{P}_{t+1}}^{(h)} \in \mathbb{R}^{(N+1)\times M}$. (Symbol–module correspondence is depicted in 2.) By proposition 1, eq. (14) and eq. (15), feature preservation requires

$$\left[\boldsymbol{S}_{\boldsymbol{X}_t\boldsymbol{X}_t}^{(h)}\boldsymbol{V}_{\boldsymbol{X}_t}^{(h)} + \boldsymbol{S}_{\boldsymbol{X}_t\boldsymbol{P}_{t+1}}^{(h)}\boldsymbol{V}_{\boldsymbol{P}_{t+1}}^{(h)}\right]_{h=1}^{H}\boldsymbol{W}_o = \left[\boldsymbol{S}_{\boldsymbol{X}_t\boldsymbol{X}_t}^{(h)}\boldsymbol{V}_{\boldsymbol{X}_t}^{(h)} + \boldsymbol{S}_{\boldsymbol{X}_t\boldsymbol{P}_t}^{(h)}\boldsymbol{V}_{\boldsymbol{P}_t}^{(h)}\right]_{h=1}^{H}\boldsymbol{W}_o \tag{18}$$

In MHSA, $\boldsymbol{W}_o$ fuses per-head outputs into a unified representation (Vaswani et al., 2017; Horn & Johnson, 2012). Since $\boldsymbol{W}_o$ may in principle be singular, we impose stronger sufficient conditions by requiring per-head equality:

$$\boldsymbol{S}_{\boldsymbol{X}_t\boldsymbol{P}_{t+1}}^{(h)}\boldsymbol{V}_{\boldsymbol{P}_{t+1}}^{(h)} - \boldsymbol{S}_{\boldsymbol{X}_t\boldsymbol{P}_t}^{(h)}\boldsymbol{V}_{\boldsymbol{P}_t}^{(h)} = \boldsymbol{0}. \tag{19}$$

Direct algebraic characterization is intractable due to the nonlinearity of softmax. Following NSP[2] (Lu et al., 2024), we instead require invariance of the attention scores:

$$\boldsymbol{A}_{\boldsymbol{Z}_{t,t}}^{(h)} = \boldsymbol{A}_{\boldsymbol{Z}_{t,t+1}}^{(h)} \Leftrightarrow \boldsymbol{Q}_{\boldsymbol{X}_t}^{(h)}\left(\boldsymbol{K}_{\boldsymbol{P}_{t+1}}^{(h)}{}^{\top} - \boldsymbol{K}_{\boldsymbol{P}_t}^{(h)}{}^{\top}\right) = \boldsymbol{0} \Rightarrow \boldsymbol{S}_{\boldsymbol{Z}_{t,t}}^{(h)} = \boldsymbol{S}_{\boldsymbol{Z}_{t,t+1}}^{(h)}. \tag{20}$$

That is, the attention scores from image tokens to prompts remain unchanged before and after the prompt update. Under eq. (20), condition eq. (19) reduces to

$$\boldsymbol{S}_{\boldsymbol{X}_t\boldsymbol{P}_t}^{(h)}\left(\boldsymbol{V}_{\boldsymbol{P}_{t+1}}^{(h)} - \boldsymbol{V}_{\boldsymbol{P}_t}^{(h)}\right) = \boldsymbol{0}. \tag{21}$$

By substituting eq. (11), we obtain per-head sufficient conditions for feature preservation:

$$\begin{cases} \boldsymbol{Q}_{\boldsymbol{X}_t}^{(h)}\left(\boldsymbol{K}_{\boldsymbol{P}_{t+1}}^{(h)}{}^{\top} - \boldsymbol{K}_{\boldsymbol{P}_t}^{(h)}{}^{\top}\right) = \boldsymbol{Q}_{\boldsymbol{X}_t}^{(h)}\boldsymbol{W}_k^{(h)}{}^{\top}\left(\mathrm{LN}\left(\boldsymbol{P}_t + \Delta\boldsymbol{P}\right) - \mathrm{LN}\left(\boldsymbol{P}_t\right)\right)^{\top} = \boldsymbol{0} \\ \boldsymbol{S}_{\boldsymbol{X}_t\boldsymbol{P}_t}^{(h)}\left(\boldsymbol{V}_{\boldsymbol{P}_{t+1}}^{(h)} - \boldsymbol{V}_{\boldsymbol{P}_t}^{(h)}\right) = \boldsymbol{S}_{\boldsymbol{X}_t\boldsymbol{P}_t}^{(h)}\left(\mathrm{LN}\left(\boldsymbol{P}_t + \Delta\boldsymbol{P}\right) - \mathrm{LN}\left(\boldsymbol{P}_t\right)\right)\boldsymbol{W}_v^{(h)} = \boldsymbol{0} \end{cases} \tag{22}$$

To reduce the conditions to direct dependence on $\Delta\boldsymbol{P}$, we expand the LayerNorm:

$$\mathrm{LN}\left(\boldsymbol{P}_t + \Delta\boldsymbol{P}\right) = \frac{\boldsymbol{P}_t + \Delta\boldsymbol{P} - \boldsymbol{\mu}_{\boldsymbol{P}_t+\Delta\boldsymbol{P}}}{\boldsymbol{\sigma}_{\boldsymbol{P}_t+\Delta\boldsymbol{P}}} \cdot \boldsymbol{\gamma} + \boldsymbol{\beta}, \tag{23}$$

where $\boldsymbol{\mu}_{\boldsymbol{P}_t+\Delta\boldsymbol{P}}, \boldsymbol{\sigma}_{\boldsymbol{P}_t+\Delta\boldsymbol{P}} \in \mathbb{R}^M$ are row-wise statistics and $\boldsymbol{\gamma}, \boldsymbol{\beta} \in \mathbb{R}^D$ are shared affine parameters (broadcast in implementation). To enable an algebraically exact characterization of prompt interactions, we explicitly reformulate LayerNorm in matrix form. A detailed derivation is provided in Appendix B. Let $\boldsymbol{C} = \boldsymbol{I}_D - \frac{1}{D}\mathbf{1}_D\mathbf{1}_D^\top (\in \mathbb{R}^{D \times D})$ be the centering matrix, $\boldsymbol{D}_{\boldsymbol{\sigma}_{\boldsymbol{P}_t+\Delta\boldsymbol{P}}} (\in \mathbb{R}^{M \times M})$ [1] the diagonal scaling matrix from $\boldsymbol{\sigma}_{\boldsymbol{P}_t+\Delta\boldsymbol{P}}$, and $\boldsymbol{\Gamma} (\in \mathbb{R}^{D \times D})$ the diagonal matrix from $\boldsymbol{\gamma}$. Then

$$\mathrm{LN}\left(\boldsymbol{P}_t + \Delta\boldsymbol{P}\right) = \boldsymbol{D}_{\boldsymbol{\sigma}_{\boldsymbol{P}_t+\Delta\boldsymbol{P}}}^{-1}\left(\boldsymbol{P}_t + \Delta\boldsymbol{P}\right)\boldsymbol{C}\boldsymbol{\Gamma} + \mathbf{1}_M\boldsymbol{\beta}^\top. \tag{24}$$

Note that pretrained $\boldsymbol{\beta}$ and $\boldsymbol{\gamma}$ are frozen. If we introduce a variance-invariance assumption

$$\boldsymbol{\sigma}_{\boldsymbol{P}_t} = \boldsymbol{\sigma}_{\boldsymbol{P}_t+\Delta\boldsymbol{P}} \triangleq \boldsymbol{\sigma}, \tag{25}$$

then

$$\mathrm{LN}\left(\boldsymbol{P}_t + \Delta\boldsymbol{P}\right) - \mathrm{LN}\left(\boldsymbol{P}_t\right) = \boldsymbol{D}_{\boldsymbol{\sigma}}^{-1}\Delta\boldsymbol{P}\boldsymbol{C}\boldsymbol{\Gamma}. \tag{26}$$

Substituting eq. (26) into eq. (22), the sufficient constraints reduce to the following linear form:

$$\begin{cases} \boldsymbol{Q}_{\boldsymbol{X}_t}^{(h)}\boldsymbol{W}_k^{(h)\top}\boldsymbol{\Gamma}\boldsymbol{C}\Delta\boldsymbol{P}^\top = \mathbf{0} \\ \boldsymbol{S}_{\boldsymbol{X}_t\boldsymbol{P}_t}^{(h)}\boldsymbol{D}_{\boldsymbol{\sigma}}^{-1}\Delta\boldsymbol{P}\boldsymbol{C}\boldsymbol{\Gamma}\boldsymbol{W}_v^{(h)} = \mathbf{0} \end{cases} \quad h = 1, 2, \ldots, H. \tag{27}$$

The first constraint ensures that, after centering and scaling, prompt updates do not perturb the scoring pattern from image tokens to prompts. The second ensures that they do not perturb the value aggregation pathway. Therefore, under the assumption eq. (25), eq. (27) provides a set of per-head sufficient conditions for feature preservation eq. (7).

## 4.3 Optimization under Feature Preservation Constraints

To enforce the variance-invariance assumption eq. (25), we adopt a soft constraint by introducing a standard deviation alignment loss:

$$\mathcal{L} = \mathcal{L}_{\mathrm{ce}} + \lambda\mathcal{L}_{\mathrm{std}}, \quad \mathcal{L}_{\mathrm{std}} = \|\boldsymbol{\sigma}_{\boldsymbol{P}_{t+1}} - \boldsymbol{\sigma}_{\boldsymbol{P}_t}\|_1, \tag{28}$$

where $\mathcal{L}$ is the total loss, $\mathcal{L}_{\mathrm{ce}}$ is the cross-entropy loss, and $\lambda$ balances the alignment term. Under this assumption, we obtain the sufficient conditions in eq. (27), constraining $\Delta\boldsymbol{P}$. We adopt a right-side nullification form on the second constraint $\boldsymbol{S}_{\boldsymbol{X}_t\boldsymbol{P}_t}^{(h)}\boldsymbol{D}_{\boldsymbol{\sigma}}^{-1}\Delta\boldsymbol{P}\boldsymbol{C}\boldsymbol{\Gamma}\boldsymbol{W}_v^{(h)} = \mathbf{0}$. The rationale is to avoid circular dependence: if $\boldsymbol{D}_{\boldsymbol{\sigma}}^{-1}$ remains on the solving side, then $\Delta\boldsymbol{P}$ becomes entangled with the current statistic $\boldsymbol{\sigma}$, complicating implementation and hurting numerical stability. In contrast, the right-sided formulation $\Delta\boldsymbol{P}\boldsymbol{C}\boldsymbol{\Gamma}\boldsymbol{W}_v^{(h)} = \mathbf{0}$ removes explicit dependence on $\boldsymbol{\sigma}$, yielding more stable behavior. Moreover, it only involves frozen pretrained parameters, reducing the computational overhead. Let $\boldsymbol{R}_k^{(h)} = \boldsymbol{C}\boldsymbol{\Gamma}\boldsymbol{W}_k^{(h)}\boldsymbol{Q}_{\boldsymbol{X}_t}^{(h)\top}$, $\boldsymbol{R}_v^{(h)} = \boldsymbol{C}\boldsymbol{\Gamma}\boldsymbol{W}_v^{(h)}$, and define

$$\boldsymbol{R} = \begin{bmatrix} \boldsymbol{R}_k^{(1)} & \cdots & \boldsymbol{R}_k^{(H)} & \boldsymbol{R}_v^{(1)} & \cdots & \boldsymbol{R}_v^{(H)} \end{bmatrix} \in \mathbb{R}^{D \times 2D}. \tag{29}$$

Our implementation enforces, at every optimization step, $\Delta\boldsymbol{P}\boldsymbol{R} = \mathbf{0}$. Equivalently, each row of $\Delta\boldsymbol{P}$ lies in the left null space of $\boldsymbol{R}$. Since the effective constraint size grows with tasks, handling $\boldsymbol{R}$ directly can be expensive. We thus use the following basic result:

**Lemma 2.** *For $\boldsymbol{A} \in \mathbb{R}^{n \times m}$ with left null space $\mathcal{N}_L\left(\boldsymbol{A}\right) = \{\boldsymbol{x}\boldsymbol{A} = \mathbf{0} \mid \boldsymbol{x}^\top \in \mathbb{R}^n\}$, it holds that $\mathcal{N}_L\left(\boldsymbol{A}\right) = \mathcal{N}_L\left(\boldsymbol{A}\boldsymbol{A}^\top\right)$.*

Proof of lemma 2 is in Appendix A. Lemma 2 implies that $\Delta\boldsymbol{P}\boldsymbol{R} = \mathbf{0}$ is equivalent to requiring each row of $\Delta\boldsymbol{P}$ to lie in $\mathcal{N}_L\left(\boldsymbol{R}\boldsymbol{R}^\top\right)$, where $\boldsymbol{R}\boldsymbol{R}^\top \in \mathbb{R}^{D \times D}$ has fixed dimension during training.

---

[1] In practice, a small constant $\epsilon > 0$ is added to the variance when computing each row's standard deviation, ensuring positivity and numerical stability. Hence all standard deviations are strictly greater than zero, and the corresponding diagonal matrix is invertible.

In practice, we perform Singular Value Decomposition (SVD) on $\boldsymbol{R}\boldsymbol{R}^\top$, collect the left singular vectors corresponding to zero singular values to form an orthogonal basis $\boldsymbol{U}_0$, so that $\mathcal{N}_L\left(\boldsymbol{R}\boldsymbol{R}^\top\right) = \mathrm{span}\left(\boldsymbol{U}_0\right)$ and its null-space projector is $\boldsymbol{B_R} = \boldsymbol{U}_0\boldsymbol{U}_0^\top$. Let $\Delta\boldsymbol{P}_{\mathrm{raw}}$ denote the raw update from back-propagation. We apply the feature-preservation projection

$$\Delta\boldsymbol{P} \leftarrow \Delta\boldsymbol{P}_{\mathrm{raw}}\boldsymbol{B_R}, \tag{30}$$

which guarantees that the feature-preservation sufficient conditions (i.e., eq. (27)) are satisfied.

### 4.4 CLASSIFICATION HEAD PRESERVATION

Proposition 2 requires the classification head to maintain its mapping of the [CLS] token across tasks. This condition is naturally fulfilled by task-specific heads, where only the current task's head is updated and all previous heads are frozen. Given CNSP's feature-preservation constraint, the representations fed to earlier heads remain invariant, so their input–output mappings are preserved automatically. At inference, we concatenate logits from all task-specific heads and predict the class with highest confidence. An illustration is provided in fig. 4, with further discussion in Appendix F.1.

**Summary.** CNSP enforces feature preservation through null-space projected updates and ensures head preservation through frozen task-specific classifiers, resulting in end-to-end representational and decision-level consistency across tasks. Detailed algorithmic descriptions are provided in Appendix C, and comparison with NSP[2] is included in Appendix D.

## 5 EXPERIMENTS

### 5.1 EXPERIMENTAL SETTINGS

**Benchmarks and Evaluation Metrics.** We evaluate CNSP on three benchmarks widely used in prompt-based continual learning: 10/20-Split CIFAR-100 (Krizhevsky et al., 2009), 10-Split ImageNet-R (Hendrycks et al., 2021), and 10-Split DomainNet (Peng et al., 2019). We also include a domain-incremental benchmark, 6-Split-DomainNet, where each task contains a single distinct visual domain. Performance is measured by Last Average Accuracy (ACC, ↑) and Forgetting (↓). Details of the benchmarks and metric definitions are provided in Appendix E.1 and E.2.

**Implementation.** Our main experiments use three backbones: ImageNet-21K pretrained ViT-B/16 (Dosovitskiy et al., 2020), DINOv2 ViT-B/14 (Oquab et al., 2023), and AugReg ViT-L/16 (Steiner et al., 2021), each equipped with four learnable prompt tokens per layer. Experiments follow the class-incremental setting with disjoint classes and unknown task identity at inference, and are trained with Adam (Kingma & Ba, 2015) using a loss that combines cross-entropy with prompt alignment regularizers (eq. (28)). Full implementation details are provided in Appendix E.3.

**Competitors.** We compare CNSP against the state-of-the-art prompt-based continual learning methods, including L2P (Wang et al., 2022b), Dual-Prompt (Wang et al., 2022a), CodaPrompt (Smith et al., 2023), CPrompt (Gao et al., 2024), EvoPrompt (Kurniawan et al., 2024), PGP (Qiao et al., 2024), LGCL (Khan et al., 2023), ConvPrompt (Roy et al., 2024), CPG (Lu et al., 2025), and NSP[2] (Lu et al., 2024).

### 5.2 MAIN RESULTS AND ANALYSIS

**Overall Comparison on ImageNet-21K ViT-B/16.** As shown in table 1, CNSP consistently outperforms prior prompt-based continual learning methods on all benchmarks using the ImageNet-21K pretrained ViT-B/16 backbone. On 10-Split DomainNet, for example, CNSP achieves a 9.7% relative reduction in forgetting ($7.33\% \rightarrow 6.62\%$) while also improving accuracy from 84.17% to 84.51% over NSP[2], indicating that enforcing principled null-space consistency yields measurable gains even in accuracy-saturated regimes. Task-wise curves (fig. 3) show the same pattern: CNSP maintains higher or comparable ACC and consistently lower forgetting across all tasks. CNSP also surpasses strong baselines including ConvPrompt and CPG, demonstrating that the proposed matrix-form analysis and right-side projection deliver robust, generalizable benefits in both within-domain and cross-domain continual learning.

Table 1: Comparison with existing prompt-based methods using ImageNet-21K pretrained backbones. Results (%) are reported as mean ± standard deviation over three runs. Results marked with †, ‡, and § are reproduced by Qiao et al. (2024), Gao et al. (2024), and Lu et al. (2025), respectively, due to the lack of official results.

| Method | 10-Split CIFAR100 | | 20-Split CIFAR100 | | 10-Split ImageNet-R | | 10-Split DomainNet | |
|---|---|---|---|---|---|---|---|---|
| | ACC(↑) | Forgetting(↓) | ACC(↑) | Forgetting(↓) | ACC(↑) | Forgetting(↓) | ACC(↑) | Forgetting(↓) |
| L2P | $83.83_{\pm0.04}$ | $7.63_{\pm0.30}$ | $81.29_{\pm0.43}{}^{\dagger}$ | $8.96_{\pm0.38}{}^{\dagger}$ | $61.57_{\pm0.66}$ | $9.73_{\pm0.47}$ | $81.17_{\pm0.83}{}^{\ddagger}$ | $8.98_{\pm1.25}{}^{\ddagger}$ |
| DualPrompt | $86.51_{\pm0.33}$ | $5.16_{\pm0.09}$ | $82.98_{\pm0.47}{}^{\dagger}$ | $8.20_{\pm0.08}{}^{\dagger}$ | $68.13_{\pm0.49}$ | $4.68_{\pm0.20}$ | $81.70_{\pm0.78}{}^{\ddagger}$ | $8.04_{\pm0.31}{}^{\ddagger}$ |
| CodaPrompt | $86.25_{\pm0.74}$ | $1.67_{\pm0.26}$ | - | - | $75.45_{\pm0.56}$ | $1.64_{\pm0.10}$ | $80.04_{\pm0.79}{}^{\ddagger}$ | $10.16_{\pm0.35}{}^{\ddagger}$ |
| CPrompt | $87.82_{\pm0.21}$ | $5.06_{\pm0.50}$ | $83.97_{\pm0.31}{}^{\S}$ | $6.85_{\pm0.43}{}^{\S}$ | $77.14_{\pm0.11}$ | $5.97_{\pm0.68}$ | $82.97_{\pm0.34}$ | $7.45_{\pm0.93}$ |
| EvoPrompt | $87.97_{\pm0.30}$ | $2.60_{\pm0.42}$ | $84.64_{\pm0.14}$ | $3.98_{\pm0.24}$ | $76.83_{\pm0.08}$ | $2.78_{\pm0.06}$ | $79.50_{\pm0.29}{}^{\S}$ | $3.81_{\pm0.36}{}^{\S}$ |
| PGP | $86.92_{\pm0.05}$ | $5.35_{\pm0.19}$ | $83.74_{\pm0.01}$ | $7.91_{\pm0.15}$ | $69.34_{\pm0.05}$ | $4.53_{\pm0.04}$ | $80.41_{\pm0.25}{}^{\S}$ | $8.39_{\pm0.18}{}^{\S}$ |
| LGCL | $87.23_{\pm0.21}$ | $5.10_{\pm0.15}$ | - | - | $69.46_{\pm0.04}$ | $4.20_{\pm0.06}$ | - | - |
| ConvPrompt | $88.87_{\pm0.33}$ | $4.75_{\pm0.15}$ | $87.22_{\pm0.42}{}^{\S}$ | $5.43_{\pm0.29}{}^{\S}$ | $77.86_{\pm0.25}$ | $4.33_{\pm0.24}$ | $79.47_{\pm0.35}{}^{\S}$ | $6.49_{\pm0.43}{}^{\S}$ |
| CPG | $90.63_{\pm0.44}$ | $3.98_{\pm0.65}$ | $88.08_{\pm0.77}$ | $5.20_{\pm0.64}$ | $78.63_{\pm0.52}$ | $7.18_{\pm0.62}$ | $83.21_{\pm0.67}$ | $7.09_{\pm0.82}$ |
| $NSP^2$ | $90.70_{\pm0.17}$ | $4.20_{\pm0.35}$ | $88.59_{\pm0.46}$ | $4.45_{\pm0.60}$ | $79.17_{\pm0.63}$ | $5.06_{\pm1.07}$ | $84.17_{\pm0.54}$ | $7.33_{\pm0.94}$ |
| **CNSP (Ours)** | **91.01**$_{\pm0.10}$ | **3.91**$_{\pm0.47}$ | **89.42**$_{\pm0.58}$ | **4.23**$_{\pm0.51}$ | **79.69**$_{\pm0.32}$ | **4.62**$_{\pm0.66}$ | **84.51**$_{\pm0.41}$ | **6.62**$_{\pm0.26}$ |

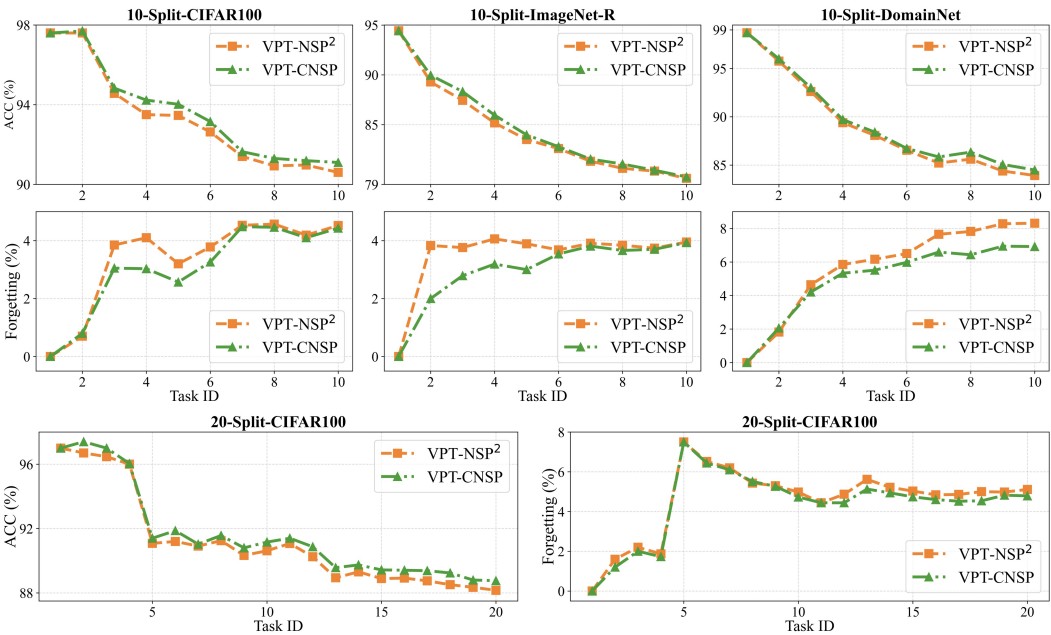

Figure 3: Task-wise performance curves on 10-Split CIFAR100, 20-Split CIFAR100, 10-Split ImageNet-R, and 10-Split DomainNet using the ViT-B/16-21K backbone. For each benchmark, results include the last average accuracy (ACC, %), where higher is better, and the corresponding forgetting (%), where lower indicates better retention of prior knowledge.

**Robustness Across Backbones.** We further evaluate CNSP on two distinct backbones—ViT-L/16 (AugReg) (Steiner et al., 2021) and the self-supervised DINOv2 ViT-B/14 (Oquab et al., 2023)—with results shown in tables 2 and 3. Across both models, CNSP consistently outperforms $NSP^2$. The improvement is especially pronounced on ViT-L/16, where approximation errors in high-dimensional token spaces accumulate, while CNSP's exact right-side projection remains stable. On 10-Split ImageNet-R, forgetting drops from 6.50% to 4.29% (34% relative reduction), alongside an ACC increase from 83.52% to 84.78%. The gains on DINOv2 further show that CNSP does not rely on supervised ImageNet-21K pretraining and transfers well to self-supervised features. Overall, these results demonstrate that CNSP is robust to variations in model scale and pretraining strategy.

Table 2: Results on four benchmarks with ViT-L/16 (AugReg, IN-21k) backbone.

| Method | 10-Split ImageNet-R | | 10-Split DomainNet | | 10-Split CIFAR100 | | 20-Split CIFAR100 | |
|---|---|---|---|---|---|---|---|---|
| | ACC($\uparrow$) | Forgetting($\downarrow$) | ACC($\uparrow$) | Forgetting($\downarrow$) | ACC($\uparrow$) | Forgetting($\downarrow$) | ACC($\uparrow$) | Forgetting($\downarrow$) |
| Upper Bound | 88.58 | 0.00 | 90.48 | 0.00 | 95.06 | 0.00 | 95.06 | 0.00 |
| Lower Bound | 79.92 | 15.51 | 80.85 | 15.88 | 88.91 | 10.11 | 87.40 | 11.32 |
| NSP$^2$ | 83.52 | 6.50 | 84.38 | 10.28 | 92.20 | 4.14 | 90.83 | 6.34 |
| **CNSP (Ours)** | **84.78** | **4.29** | **85.63** | **8.03** | **92.45** | **4.01** | **91.02** | **5.24** |

Table 3: Results on four benchmarks with DINOv2 ViT-B/14 backbone.

| Method | 10-Split ImageNet-R | | 10-Split DomainNet | | 10-Split CIFAR100 | | 20-Split CIFAR100 | |
|---|---|---|---|---|---|---|---|---|
| | ACC($\uparrow$) | Forgetting($\downarrow$) | ACC($\uparrow$) | Forgetting($\downarrow$) | ACC($\uparrow$) | Forgetting($\downarrow$) | ACC($\uparrow$) | Forgetting($\downarrow$) |
| Upper Bound | 89.58 | 0.00 | 89.65 | 0.00 | 93.22 | 0.00 | 93.22 | 0.00 |
| Lower Bound | 78.42 | 17.66 | 68.87 | 31.89 | 78.01 | 22.27 | 77.84 | 21.80 |
| NSP$^2$ | 85.22 | 4.91 | 84.52 | 10.06 | 89.43 | 5.08 | 87.15 | 5.92 |
| **CNSP (Ours)** | **85.67** | **4.49** | **85.27** | **8.50** | **89.87** | **4.79** | **87.29** | **5.72** |

**Robustness to Domain Shifts.** To evaluate robustness under task heterogeneity, we test CNSP on 6-Split DomainNet, where each task corresponds to a distinct visual domain. As shown in table 4, forgetting decreases from 6.67% to 5.84% on run 1 (a 12% relative reduction) with an ACC gain (83.55% $\rightarrow$ 84.32%). These results indicate that CNSP maintains stable performance under substantial domain shifts without relying on assumptions about invariant prompt distributions.

Table 4: Results on the 6-Split DomainNet benchmark across three runs with ViT-B/16-21K.

| Method | Run 1 | | Run 2 | | Run 3 | |
|---|---|---|---|---|---|---|
| | ACC($\uparrow$) | Forgetting($\downarrow$) | ACC($\uparrow$) | Forgetting($\downarrow$) | ACC($\uparrow$) | Forgetting($\downarrow$) |
| Upper Bound | 91.68 | 0.00 | 91.94 | 0.00 | 91.88 | 0.00 |
| Lower Bound | 63.34 | 34.11 | 79.17 | 18.52 | 78.22 | 17.65 |
| NSP$^2$ | 83.55 | 6.67 | 84.85 | 7.91 | 84.76 | 3.36 |
| **CNSP (Ours)** | **84.32** | **5.84** | **85.34** | **6.44** | **85.28** | **3.07** |

**Ablation Study.** A detailed ablation study is provided in Appendix E.5. In practice, it shows that the right-side null-space projection is the primary driver of CNSP's performance, while the variance-preservation loss mainly contributes to training stability.

**Summary.** CNSP delivers consistent gains across model scales, pretraining paradigms, and domain-shift scenarios. Additional analysis on prompt expressive capacity is provided in Appendix E.6.

# 6 CONCLUSION

We introduced CNSP, a principled framework that provides the first end-to-end theoretical formulation of prompt-based continual learning. By modeling MHSA, LayerNorm, and classification heads in exact matrix form, CNSP derives unified sufficient consistency conditions for joint feature and head preservation, leading directly to a simple and tractable projection rule. Across diverse benchmarks and Transformer backbones—including large-scale, cross-domain, and self-supervised settings—CNSP consistently improves accuracy and reduces forgetting. Looking forward, CNSP opens several promising directions, including theoretically grounded head designs, extensions to multimodal continual learning, adaptive prompt allocation, and tighter analyses of nonlinear components such as softmax. Additional discussions and limitations are provided in Appendix F.

REPRODUCIBILITY STATEMENT

Our experiments are conducted on publicly available datasets (e.g., CIFAR-100, ImageNet). We report the details of our training setup, including hyperparameters (batch size, learning rate, optimizer), number of epochs, and evaluation metrics, in Appendix E. To ensure reproducibility, we will release the complete source code and training/evaluation scripts upon acceptance of the paper.

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

# APPENDIX

## A  PROOFS OF LEMMAS AND PROPOSITIONS

**Lemma 1.** *Consider the computation of LayerNorm* $\mathrm{LN}(\cdot)$. *Let* $\boldsymbol{A} = [\boldsymbol{A}^{(1)}; \boldsymbol{A}^{(2)}]$ *with* $\boldsymbol{A}^{(1)} \in \mathbb{R}^{n \times d}$ *and* $\boldsymbol{A}^{(2)} \in \mathbb{R}^{m \times d}$. *Then* $\mathrm{LN}(\boldsymbol{A}) = [\mathrm{LN}(\boldsymbol{A}^{(1)}); \mathrm{LN}(\boldsymbol{A}^{(2)})]$.

*Proof.* Expanding the LayerNorm operations gives

$$\mathrm{LN}(\boldsymbol{A}) = \frac{\boldsymbol{A} - \boldsymbol{\mu_A}}{\boldsymbol{\sigma_A}} \cdot \boldsymbol{\gamma} + \boldsymbol{\beta}, \quad \boldsymbol{\mu_A}, \boldsymbol{\sigma_A} \in \mathbb{R}^{n+m}, \quad \boldsymbol{\gamma}, \boldsymbol{\beta} \in \mathbb{R}^d, \tag{31}$$

where broadcasting is applied to match the dimensions $(n+m) \times d$. By matrix reformulation of the broadcast, let

$$\boldsymbol{C} = \boldsymbol{I}_d - \frac{1}{d}\boldsymbol{1}_d\boldsymbol{1}_d^\top \in \mathbb{R}^{d \times d}, \quad \boldsymbol{D}_{\boldsymbol{\sigma_A}} = \mathrm{diag}(\boldsymbol{\sigma_A}) \in \mathbb{R}^{(n+m) \times (n+m)}, \quad \boldsymbol{\Gamma} = \mathrm{diag}(\boldsymbol{\gamma}) \in \mathbb{R}^{d \times d}. \tag{32}$$

Then

$$\mathrm{LN}(\boldsymbol{A}) = \boldsymbol{D}_{\boldsymbol{\sigma_A}}^{-1}\boldsymbol{A}\boldsymbol{C}\boldsymbol{\Gamma} + \boldsymbol{1}_{n+m}\boldsymbol{\beta}^\top. \tag{33}$$

In practice, the standard deviation of the $i$-th row is computed as

$$\sqrt{\frac{1}{d}\mathrm{Var}(\boldsymbol{A}[i]) + \epsilon}, \tag{34}$$

where $\mathrm{Var}(\cdot)$ is the operation to compute variance, $\epsilon > 0$ ensures non-degeneracy. Hence, $\boldsymbol{D}_{\boldsymbol{\sigma_A}}$ is invertible. Since $\boldsymbol{\sigma_A}$ corresponds row-wise to $\boldsymbol{A}$, we may write

$$\boldsymbol{\sigma_A} = \begin{bmatrix} \boldsymbol{\sigma_{A^{(1)}}} & \boldsymbol{0} \\ \boldsymbol{0} & \boldsymbol{\sigma_{A^{(2)}}} \end{bmatrix}, \quad \boldsymbol{D}_{\boldsymbol{\sigma_A}} = \begin{bmatrix} \boldsymbol{D}_{\boldsymbol{\sigma_{A^{(1)}}}} & \boldsymbol{0} \\ \boldsymbol{0} & \boldsymbol{D}_{\boldsymbol{\sigma_{A^{(2)}}}} \end{bmatrix}. \tag{35}$$

Therefore,

$$\mathrm{LN}(\boldsymbol{A}) = \begin{bmatrix} \boldsymbol{D}_{\boldsymbol{\sigma_{A^{(1)}}}}\boldsymbol{A}^{(1)}\boldsymbol{C}\boldsymbol{\Gamma} + \boldsymbol{1}_n\boldsymbol{\beta}^\top \\ \boldsymbol{D}_{\boldsymbol{\sigma_{A^{(2)}}}}\boldsymbol{A}^{(2)}\boldsymbol{C}\boldsymbol{\Gamma} + \boldsymbol{1}_m\boldsymbol{\beta}^\top \end{bmatrix} = \begin{bmatrix} \mathrm{LN}(\boldsymbol{A}^{(1)}) \\ \mathrm{LN}(\boldsymbol{A}^{(2)}) \end{bmatrix}. \tag{36}$$

$\square$

**Lemma 2.** *For* $\boldsymbol{A} \in \mathbb{R}^{n \times m}$ *with left null space* $\mathcal{N}_L(\boldsymbol{A}) = \{\boldsymbol{x}\boldsymbol{A} = \boldsymbol{0} \mid \boldsymbol{x}^\top \in \mathbb{R}^n\}$, *it holds that* $\mathcal{N}_L(\boldsymbol{A}) = \mathcal{N}_L(\boldsymbol{A}\boldsymbol{A}^\top)$.

*Proof.* $\forall \boldsymbol{x} \in \mathcal{N}_L(\boldsymbol{A})$, we have $\boldsymbol{x}\boldsymbol{A} = \boldsymbol{0}$, which implies $\boldsymbol{x}\boldsymbol{A}\boldsymbol{A}^\top = \boldsymbol{0}$, i.e., $\boldsymbol{x} \in \mathcal{N}_L(\boldsymbol{A}\boldsymbol{A}^\top)$. Conversely, $\forall \boldsymbol{x} \in \mathcal{N}_L(\boldsymbol{A}\boldsymbol{A}^\top)$, we have $\boldsymbol{x}\boldsymbol{A}\boldsymbol{A}^\top = \boldsymbol{0}$. Then, $0 = \boldsymbol{x}\boldsymbol{A}\boldsymbol{A}^\top\boldsymbol{x}^\top = \|\boldsymbol{A}^\top\boldsymbol{x}^\top\|^2$, which implies $\boldsymbol{x}\boldsymbol{A} = \boldsymbol{0}$, i.e., $\boldsymbol{x} \in \mathcal{N}_L(\boldsymbol{A})$. $\square$

**Lemma 3.** *Consider the operation* $\mathrm{MLP}(\cdot)$ *in a standard ViT MLP (Dosovitskiy et al., 2020), containing two layers with a GELU non-linearity, defined as a linear projection followed by a GELU activation and another linear projection. For a block matrix* $\boldsymbol{A} = [\boldsymbol{A}^{(1)}; \boldsymbol{A}^{(2)}]$, *where* $\boldsymbol{A}^{(1)} \in \mathbb{R}^{n \times d}$ *and* $\boldsymbol{A}^{(2)} \in \mathbb{R}^{m \times d}$, *we have*

$$\mathrm{MLP}(\boldsymbol{A}) = \begin{bmatrix} \mathrm{MLP}(\boldsymbol{A}^{(1)}) \\ \mathrm{MLP}(\boldsymbol{A}^{(2)}) \end{bmatrix} \tag{37}$$

*Proof.* By definition,

$$\mathrm{MLP}(\boldsymbol{A}) = \phi(\boldsymbol{A}\boldsymbol{W}_1 + \boldsymbol{1}_{n+m}\boldsymbol{b}_1^\top)\boldsymbol{W}_2 + \boldsymbol{1}_{n+m}\boldsymbol{b}_2^\top, \tag{38}$$

where $\boldsymbol{W}_1 \in \mathbb{R}^{d \times d_h}$, $\boldsymbol{W}_2 \in \mathbb{R}^{d_h \times d}$ are weight matrices of two linear projections respectively, $\boldsymbol{b}_1 \in \mathbb{R}^{d_h}$, $\boldsymbol{b}_2 \in \mathbb{R}^d$ are bias vectors, and $\phi$ denotes the element-wise GELU.

Since linear transformations, element-wise nonlinearities, and bias additions act independently on each row, the block structure is preserved. Explicitly,

$$\phi(\boldsymbol{A}\boldsymbol{W}_1 + \boldsymbol{1}_{n+m}\boldsymbol{b}_1^\top) = \phi\left(\begin{bmatrix} \boldsymbol{A}^{(1)} \\ \boldsymbol{A}^{(2)} \end{bmatrix}\boldsymbol{W}_1 + \begin{bmatrix} \boldsymbol{1}_n \\ \boldsymbol{1}_m \end{bmatrix}\boldsymbol{\beta}_1^\top\right) = \begin{bmatrix} \phi(\boldsymbol{A}^{(1)}\boldsymbol{W}_1 + \boldsymbol{1}_n\boldsymbol{b}_1^\top) \\ \phi(\boldsymbol{A}^{(2)}\boldsymbol{W}_1 + \boldsymbol{1}_m\boldsymbol{b}_1^\top) \end{bmatrix}. \tag{39}$$

Similarly, multiplying by $\boldsymbol{W}_2$ and adding bias yields

$$\phi\left(\boldsymbol{A}\boldsymbol{W}_1 + \mathbf{1}_{n+m}\boldsymbol{b}_1^\top\right)\boldsymbol{W}_2 + \mathbf{1}_{n+m}\boldsymbol{b}_2^\top = \begin{bmatrix} \phi\left(\boldsymbol{A}^{(1)}\boldsymbol{W}_1 + \mathbf{1}_n\boldsymbol{b}_1^\top\right)\boldsymbol{W}_2 + \mathbf{1}_n\boldsymbol{b}_2^\top \\ \phi\left(\boldsymbol{A}^{(2)}\boldsymbol{W}_1 + \mathbf{1}_m\boldsymbol{b}_1^\top\right)\boldsymbol{W}_2 + \mathbf{1}_m\boldsymbol{b}_2^\top \end{bmatrix}, \tag{40}$$

which implies that

$$\mathrm{MLP}\left(\boldsymbol{A}\right) = \begin{bmatrix} \mathrm{MLP}\left(\boldsymbol{A}^{(1)}\right) \\ \mathrm{MLP}\left(\boldsymbol{A}^{(2)}\right) \end{bmatrix} \tag{41}$$

$\square$

**Proposition 1.** *Consider two tasks $\mathcal{T}_t$ and $\mathcal{T}_{t+1}$. If the image-token part of attention outputs remains unchanged for all blocks in VPT, i.e.,*

$$\boldsymbol{O}^{(l)}_{\boldsymbol{Z}_{t,t+1}}[: N+1] = \boldsymbol{O}^{(l)}_{\boldsymbol{Z}_{t,t}}[: N+1], \quad l = 1, 2, \ldots, L, \tag{6}$$

*then the image-token output of the final block is preserved:*

$$\boldsymbol{E}^{(L)}_{\boldsymbol{Z}_{t,t+1}}[: N+1] = \boldsymbol{E}^{(L)}_{\boldsymbol{Z}_{t,t}}[: N+1], \tag{7}$$

*which we refer to as **feature preservation**. Here $\boldsymbol{O}^{(l)}_{\boldsymbol{Z}_{t,t+1}}$, $\boldsymbol{O}^{(l)}_{\boldsymbol{Z}_{t,t}}$, $\boldsymbol{E}^{(L)}_{\boldsymbol{Z}_{t,t+1}}$, and $\boldsymbol{E}^{(L)}_{\boldsymbol{Z}_{t,t}}$ are computed by eq. (1). In particular, the final [CLS] token is preserved:*

$$\boldsymbol{E}^{(L)}_{\boldsymbol{Z}_{t,t+1}}[: 1] = \boldsymbol{E}^{(L)}_{\boldsymbol{Z}_{t,t}}[: 1]. \tag{8}$$

*Proof.* Without loss of generality, we first omit the task indices and focus on the $l$-th transformer block. As shown in fig. 1, within a block, $\boldsymbol{Z}^{(l)}$ is added to the output $\boldsymbol{O}^{(l)}_{\boldsymbol{Z}}$ via a residual connection, followed by LayerNorm and an MLP with an additional residual connection:

$$\boldsymbol{B}^{(l)}_{\boldsymbol{Z}} = \boldsymbol{Z}^{(l)} + \boldsymbol{O}^{(l)}_{\boldsymbol{Z}}, \quad \boldsymbol{E}^{(l)}_{\boldsymbol{Z}} = \boldsymbol{B}^{(l)}_{\boldsymbol{Z}} + \mathrm{MLP}\left(\mathrm{LN}\left(\boldsymbol{B}^{(l)}_{\boldsymbol{Z}}\right)\right). \tag{42}$$

Since

$$\boldsymbol{E}^{(l)}_{\boldsymbol{Z}}[: N+1] = \boldsymbol{B}^{(l)}_{\boldsymbol{Z}}[: N+1] + \mathrm{MLP}\left(\mathrm{LN}\left(\boldsymbol{B}^{(l)}_{\boldsymbol{Z}}\right)\right)[: N+1], \tag{43}$$

by lemmas 1 and 3, we obtain

$$\mathrm{MLP}\left(\mathrm{LN}\left(\boldsymbol{B}^{(l)}_{\boldsymbol{Z}}\right)\right)[: N+1] = \mathrm{MLP}\left(\mathrm{LN}\left(\boldsymbol{B}^{(l)}_{\boldsymbol{Z}}[: N+1]\right)\right). \tag{44}$$

Moreover,

$$\boldsymbol{B}^{(l)}_{\boldsymbol{Z}}[: N+1] = \boldsymbol{Z}^{(l)}[: N+1] + \boldsymbol{O}^{(l)}_{\boldsymbol{Z}}[: N+1] = \boldsymbol{X}^{(l)} + \boldsymbol{O}^{(l)}_{\boldsymbol{Z}}[: N+1], \tag{45}$$

thus,

$$\boldsymbol{E}^{(l)}_{\boldsymbol{Z}}[: N+1] = \boldsymbol{X}^{(l)} + \boldsymbol{O}^{(l)}_{\boldsymbol{Z}}[: N+1] + \mathrm{MLP}\left(\mathrm{LN}\left(\boldsymbol{X}^{(l)} + \boldsymbol{O}^{(l)}_{\boldsymbol{Z}}[: N+1]\right)\right). \tag{46}$$

In VPT, before passing to the next block , the prompt embeddings are discarded and replaced by new learnable prompts:

$$\boldsymbol{X}^{(l+1)} = \boldsymbol{E}^{(l)}_{\boldsymbol{Z}}[: N+1], \quad \boldsymbol{Z}^{(l+1)} = [\boldsymbol{X}^{(l+1)}; \boldsymbol{P}^{(l+1)}]. \tag{47}$$

Now consider two tasks $\mathcal{T}_t$ and $\mathcal{T}_{t+1}$. We prove the statement by induction on $l$:

(i) When $l = 1$, by assumption and setting, we have

$$\boldsymbol{O}^{(1)}_{\boldsymbol{Z}_{t,t+1}}[: N+1] = \boldsymbol{O}^{(1)}_{\boldsymbol{Z}_{t,t}}[: N+1], \quad \boldsymbol{X}^{(1)}_{t,t+1} = \boldsymbol{X}^{(1)}_{t,t} = \boldsymbol{X}_t, \tag{48}$$

where $\boldsymbol{X}_t$ is the input image tokens of VPT from task $\mathcal{T}_t$ dataset. According to eq. (46), it follows that

$$\boldsymbol{E}^{(1)}_{\boldsymbol{Z}_{t,t+1}}[: N+1] = \boldsymbol{E}^{(l)}_{\boldsymbol{Z}_{t,t}}[: N+1], \quad \boldsymbol{X}^{(2)}_{t,t+1} = \boldsymbol{X}^{(2)}_{t,t}. \tag{49}$$

(ii) Suppose that for some $l = k(< L)$,

$$\boldsymbol{E}^{(k)}_{\boldsymbol{Z}_{t,t+1}}[: N+1] = \boldsymbol{E}^{(k)}_{\boldsymbol{Z}_{t,t}}[: N+1], \tag{50}$$

which implies

$$\boldsymbol{X}_{t,t+1}^{(k+1)} = \boldsymbol{X}_{t,t}^{(k+1)}. \tag{51}$$

For $l = k + 1$, we know

$$\boldsymbol{O}_{\boldsymbol{Z}_{t,t+1}}^{(k+1)}[: N + 1] = \boldsymbol{O}_{\boldsymbol{Z}_{t,t}}^{(k+1)}[: N + 1]. \tag{52}$$

By eqs. (46) and (51), it follows that

$$\boldsymbol{E}_{\boldsymbol{Z}_{t,t+1}}^{(k+1)}[: N + 1] = \boldsymbol{E}_{\boldsymbol{Z}_{t,t}}^{(k+1)}[: N + 1]. \tag{53}$$

(iii) By induction, for all $l = 1, 2, \ldots, L$, we have

$$\boldsymbol{E}_{\boldsymbol{Z}_{t,t+1}}^{(l)}[: N + 1] = \boldsymbol{E}_{\boldsymbol{Z}_{t,t}}^{(l)}[: N + 1]. \tag{54}$$

Especially, the final [CLS] token fed into the classification head is unchanged, i.e.,

$$\boldsymbol{E}_{\boldsymbol{Z}_{t,t+1}}^{(L)}[: 1] = \boldsymbol{E}_{\boldsymbol{Z}_{t,t}}^{(L)}[: 1]. \tag{55}$$

$\square$

## B  MATRIX-FORM CHARACTERIZATION OF LAYERNORM

LayerNorm operates row-wise on tokens and normalizes each embedding using its own mean and standard deviation. While this definition is straightforward in implementation, the usual broadcasting notation obscures the algebraic structure of the operation and complicates the analysis of consistency conditions. We therefore present an exact matrix-form characterization that is exactly equivalent to the standard implementation and preserves its semantics.

Let $\boldsymbol{A} \in \mathbb{R}^{n \times d}$ be a sequence of token embeddings. Standard LayerNorm is defined as

$$\mathrm{LN}\,(\boldsymbol{A}) = \frac{\boldsymbol{A} - \boldsymbol{\mu_A}}{\boldsymbol{\sigma_A}} \cdot \boldsymbol{\gamma} + \boldsymbol{\beta}, \quad \boldsymbol{\mu_A}, \boldsymbol{\sigma_A} \in \mathbb{R}^n, \quad \boldsymbol{\gamma}, \boldsymbol{\beta} \in \mathbb{R}^d, \tag{56}$$

where $\boldsymbol{\mu_A}, \boldsymbol{\sigma_A}$ are row-wise statistics and $\boldsymbol{\gamma}, \boldsymbol{\beta}$ are the learned affine parameters applied element-wise along the feature dimension.

**Exact representation of mean subtraction.** Introduce the centering matrix

$$\boldsymbol{C} = \boldsymbol{I}_d - \frac{1}{d}\boldsymbol{1}_d\boldsymbol{1}_d^\top \in \mathbb{R}^{d \times d}, \tag{57}$$

which yields

$$(\boldsymbol{A}\boldsymbol{C})[i, j] = \boldsymbol{A}[i, j] - (\frac{1}{d}\boldsymbol{A}\boldsymbol{1}_d\boldsymbol{1}_d^\top)[i, j] = \boldsymbol{A}[i, j] - \frac{1}{d}\sum_{k=1}^{d}\boldsymbol{A}[i, k] = \boldsymbol{A}[i, j] - \boldsymbol{\mu_A}[i]. \tag{58}$$

Thus, the mean subtraction step is absorbed into the linear operator $\boldsymbol{C}$ and no information is lost.

**Exact representation of variance scaling.** Define the diagonal matrix

$$\boldsymbol{D}_{\boldsymbol{\sigma_A}} = \mathrm{diag}(\boldsymbol{\sigma_A}) \in \mathbb{R}^{n \times n}. \tag{59}$$

Left multiplication by $\boldsymbol{D}_{\boldsymbol{\sigma_A}}^{-1}$ implements the row-wise division:

$$(\boldsymbol{D}_{\boldsymbol{\sigma_A}}^{-1}\boldsymbol{A})[i, j] = \frac{\boldsymbol{A}[i, j]}{\boldsymbol{\sigma_A}[i]}. \tag{60}$$

(The addition of a small positive constant in LayerNorm ensures invertibility of $\boldsymbol{D}_{\boldsymbol{\sigma_A}}$.) Therefore, the standard deviation division is exactly realized by the linear operator $\boldsymbol{D}_{\boldsymbol{\sigma_A}}^{-1}$.

**Affine transformation in matrix form.** Define

$$\boldsymbol{\Gamma} = \mathrm{diag}(\boldsymbol{\gamma}) \in \mathbb{R}^{d \times d}, \quad \boldsymbol{1}_n\boldsymbol{\beta}^\top \in \mathbb{R}^{n \times d}. \tag{61}$$

Then for every $i, j$:

$$(\boldsymbol{A}\boldsymbol{\Gamma})[i, j] = \sum_{k=1}^{d}\boldsymbol{A}[i, k]\boldsymbol{\Gamma}[k, j] = \boldsymbol{A}[i, j]\boldsymbol{\gamma}[j], \tag{62}$$

showing that right-multiplication with $\mathbf{\Gamma}$ is exactly equivalent to the broadcasted element-wise affine transformation.

**Complete matrix-form LayerNorm.** The LayerNorm operation can thus be written in exact matrix form:

$$\mathrm{LN}\left(\boldsymbol{A}\right) = \boldsymbol{D}_{\boldsymbol{\sigma}_A}^{-1}\boldsymbol{A}\boldsymbol{C}\boldsymbol{\Gamma} + \mathbf{1}_n\boldsymbol{\beta}^{\top}. \tag{63}$$

This representation strictly preserves the semantics of standard LayerNorm while revealing its linear–affine structure and removing the ambiguity of implicit broadcasting, and it enables the consistency constraints in Sec. 4 to be derived in closed algebraic form.

## C    CONSISTENT NULL-SPACE PROJECTION IN PRACTICE

In the main text (section 4.3), we derived sufficient conditions for feature preservation, requiring that each prompt update $\Delta\boldsymbol{P}$ lies in the left null space of the constraint matrix $\boldsymbol{R}$. Theoretically, this is enforced by projecting $\Delta\boldsymbol{P}$ onto the null space of $\boldsymbol{R}\boldsymbol{R}^{\top}$ (see eq. (30)). However, in practice, due to finite-precision arithmetic, exact zero singular values rarely occur, making it nontrivial to determine the effective nullity. To address this issue, following Lu et al. (2024), we estimate the nullity of $\boldsymbol{R}\boldsymbol{R}^{\top}$ using an *adaptive thresholding criterion* based on the inflection point of the singular value spectrum. Specifically, we compute the discrete second derivative of sorted singular values and use its maximizer to determine the cut-off between the null and non-null components. Formally, let $\lambda_j$ denote the $j$-th singular value of $\boldsymbol{R}\boldsymbol{R}^{\top}$ (sorted in descending order). The adaptive estimate of the nullity is given by

$$\mathrm{Nullity}\left(\boldsymbol{R}\boldsymbol{R}^{\top}\right) = D - \arg\max_{j}\left\{\lambda_{j+1} - 2\lambda_j + \lambda_{j-1}\right\}_{j=2}^{D-1}, \tag{64}$$

where the criterion detects the point of maximum curvature change in the spectrum. Empirically, this approach has proven more robust than fixed thresholds in VPT Lu et al. (2024), providing a numerically stable approximation of the null space dimension.

With the estimated null space, we then construct an orthogonal basis $\boldsymbol{U}_0$ for $\mathcal{N}_L(\boldsymbol{R}\boldsymbol{R}^{\top})$ and form the projection matrix $\boldsymbol{B}_{\boldsymbol{R}} = \boldsymbol{U}_0\boldsymbol{U}_0^{\top}$. During training, the raw prompt update $\Delta\boldsymbol{P}_{\mathrm{raw}}$ from backpropagation is consistently projected onto this null space, ensuring that the sufficient conditions for feature preservation are satisfied at every optimization step. This procedure constitutes our *Consistent Null-Space Projection (CNSP)* mechanism.

The detailed training algorithm is summarized in Algorithm 1, and the computation of the null-space projection matrix is given in Algorithm 2.

## D    COMPARISON AND DISCUSSION WITH NSP[2]

CNSP and NSP[2] both rely on projection-based updates for prompt tuning in Transformers, yet they differ substantially in how thoroughly they characterize and enforce representational consistency. Below, we highlight the key distinctions in a concise and analytically oriented manner.

**Multi-head Derivation.** NSP[2] first derives sufficient conditions under a single-head setting and then extends them to multi-head attention by direct concatenation, which lacks a full derivation and leaves potential inter-head interactions unexamined. CNSP provides a complete multi-head analysis by explicitly unfolding the MHSA forward computation. This yields rigorous per-head sufficient conditions and clarifies the role of the shared output projection $\boldsymbol{W}_o$. The resulting formulation precisely characterizes when head-wise invariance guarantees MHSA-level representational preservation, offering a fully grounded multi-head theory.

**LayerNorm Modeling and Distributional Assumptions.** NSP[2] treats LayerNorm's centering and scaling operations as cancelable scalar factors, simplifying its algebraic structure at the cost of omitting key broadcasted operations. CNSP introduces an explicit matrix-form characterization of LayerNorm, including:

- the centering matrix $\boldsymbol{C}$,
- the per-sample normalization matrix $\boldsymbol{D}_{\boldsymbol{\sigma}}^{-1}$,

---

**Algorithm 1** CNSP for VPT in Continual Learning

---

**Inputs:** Datasets $\mathcal{D}_t = \{(\boldsymbol{x}_t^{(i)}, y_t^{(i)})\}_{i=1}^{|\mathcal{T}_t|}$ for task $\mathcal{T}_t \in \{\mathcal{T}_1, \mathcal{T}_2, \dots\}$; ViT model $f_{\text{ViT}}(\cdot \mid \{\boldsymbol{P}_t^{(l)}\}_{l=1}^L)$
    with the prompts $\{\boldsymbol{P}_t^{(l)}\}_{l=1}^L$ to be optimized (the updates of classification head is omitted for
    simplicity) ; the gram matrix of constraints $\boldsymbol{G}$; the projection matrix $\boldsymbol{B_R}$

**Outputs:** Optimized prompts $\boldsymbol{P}_t^{(l)}$

1: **Initialization:** Randomly initialize $\boldsymbol{P}_t^{(l)}$; set $\boldsymbol{G}^{(l)} = \boldsymbol{0}$, $\boldsymbol{B}_R^{(l)} = \boldsymbol{I}$
2: **for** task $\mathcal{T}_t \in \{\mathcal{T}_1, \mathcal{T}_2, \dots\}$ **do**
3:     **repeat**
4:         Sample a mini-batch $\boldsymbol{x}_t, y_t \sim \mathcal{D}_t$
5:         $\hat{y}_t \leftarrow f_{\text{ViT}}(\boldsymbol{x}_t \mid \{\boldsymbol{P}_t^{(l)}\}_{l=1}^L)$
6:         $\mathcal{L}_{\text{total}} \leftarrow \text{CrossEntropy}(\hat{y}_t, y_t)$                       ▷ the classification loss
7:         **if** $t > 1$ **then**
8:             Compute $\mathcal{L}_{\text{std}}$ by eq. (28)               ▷ loss of prompts standard deviation
9:             $\mathcal{L}_{\text{total}} \leftarrow \mathcal{L}_{\text{total}} + \mathcal{L}_{\text{std}}$
10:        **end if**
11:       Get raw prompts update $\Delta \boldsymbol{P}_{\text{raw}}^{(l)}$ from optimizer using $\mathcal{L}_{\text{total}}$
12:       **if** $t > 1$ **then**
13:          $\Delta \boldsymbol{P}^{(l)} \leftarrow \Delta \boldsymbol{P}_{\text{raw}}^{(l)} \boldsymbol{B}_R^{(l)}$             ▷ consistent null-space projection, eq. (30)
14:       **else**
15:          $\Delta \boldsymbol{P}^{(l)} \leftarrow \boldsymbol{P}_{\text{raw}}^{(l)}$
16:       **end if**
17:       $\boldsymbol{P}_t^{(l)} \leftarrow \boldsymbol{P}_t^{(l)} - lr \times \Delta \boldsymbol{P}^{(l)}$
18:     **until** convergence
19:     **if** $t = 1$ **then**
20:       $\boldsymbol{R}_v^{(l)} \leftarrow \boldsymbol{C}\boldsymbol{\Gamma}^{(l)}\boldsymbol{W}_v^{(l)}$                  ▷ cache the second constraint matrix
21:     **end if**
22:     **for** $\boldsymbol{x}_t^{(i)} \in \mathcal{D}_t$ **do**
23:       $\boldsymbol{R}_k^{(l)} \leftarrow \boldsymbol{C}\boldsymbol{\Gamma}^{(l)}\boldsymbol{W}_k^{(l)}\boldsymbol{Q}_{\boldsymbol{X}_t^{(i)}}^{(l)\top}$       ▷ by forward propagation $f_{\text{ViT}}(\boldsymbol{x}_t^{(i)} \mid \{\boldsymbol{P}_t^{(l)}\}_{l=1}^L)$
24:       $\boldsymbol{G}^{(l)} \leftarrow \boldsymbol{G}^{(l)} + \boldsymbol{R}_k^{(l)}\boldsymbol{R}_k^{(l)\top} + \boldsymbol{R}_v^{(l)}\boldsymbol{R}_v^{(l)\top}$
25:     **end for**
26:     $\boldsymbol{B}_R^{(l)} \leftarrow \text{ComputingNullSpaceProjection}(\boldsymbol{G}^{(l)})$             ▷ using algorithm 2
27: **end for**

---

**Algorithm 2** Computing Null-Space Projection Matrix

---

**Inputs:** Gram matrix $\boldsymbol{G}$
**Outputs:** Null-space projection matrix $\boldsymbol{B_R}$

1: $\boldsymbol{U}, \boldsymbol{\Sigma}, \_ \leftarrow \text{SVD}(\boldsymbol{G})$                ▷ sorted by the singular values in descending order
2: Computing the nullity $N_{\boldsymbol{G}}$ of $\boldsymbol{G}$ by eq. (64)
3: $\boldsymbol{U}_0 \leftarrow \boldsymbol{U}[D - N_{\boldsymbol{G}} : D]$          ▷ get the left singular vectors of zero singular values
4: $\boldsymbol{B_R} \leftarrow \frac{\boldsymbol{U}_0 \boldsymbol{U}_0^\top}{\|\boldsymbol{U}_0 \boldsymbol{U}_0^\top\|_F}$          ▷ improve numerical stability by its Frobenius norm

---

    • the affine scaling matrix $\boldsymbol{\Gamma}$.

This exact formulation captures the full behavior of LayerNorm and enables a variance-only consistency condition, relaxing the stronger mean–variance assumption employed in NSP². The relaxation improves stability while maintaining theoretical correctness.

**Implementation of Sufficient Conditions.** A key practical distinction lies in how the two methods translate theoretical constraints into implementable projections.

NSP² requires two SVDs per layer: (i) a left projection: $M \times M$ SVD (where $M$ is the number of prompt tokens), and (ii) a right projection: $D \times D$ SVD (where $D$ is the feature dimension). CNSP's

unified derivation leads to a single right-side $D \times D$ projection, significantly reducing computational cost and memory overhead.

Because CNSP models LayerNorm exactly, the resulting constraints naturally admit a right-sided nullification form that avoids circular dependencies on statistics such as $\boldsymbol{D}_{\sigma}^{(-1)}$. This makes the projection stable across tasks and smoothly scalable to larger backbones (e.g., ViT-L/16), as observed in our experiments.

**Classification Head Preservation.** NSP$^2$ analyzes invariance within intermediate layers but does not address how classification heads interact with updated prompts. CNSP extends representational consistency to the end-to-end pipeline by deriving sufficient conditions for preserving both the backbone features (MHSA + LayerNorm) and the classification head. This provides the first mathematical characterization of when task-specific heads remain valid after prompt updates and how they decompose from feature preservation constraints.

**Summary.** Overall, NSP$^2$ provides a valuable and influential starting point for principled prompt-based continual learning. CNSP addresses the theoretical gaps in NSP$^2$ by supplying rigorous multi-head derivations, an exact LayerNorm formulation, relaxed distributional assumptions, and the first end-to-end consistency analysis. Its unified projection rule results in a more stable and scalable implementation that preserves task performance while maintaining theoretical soundness.

# E  EXPERIMENTS

## E.1  DATASETS AND BENCHMARKS

To thoroughly evaluate the proposed CNSP method in prompt-based continual learning scenarios, we conduct experiments on five continual learning benchmarks spanning both class-incremental and domain-incremental settings:

- **10/20-Split CIFAR-100** (Krizhevsky et al., 2009): CIFAR-100 consists of 60,000 $32 \times 32$ color images across 100 classes, with 500 training images and 100 test images per class. Following common practice, it is randomly partitioned into 10 or 20 tasks, where each task contains 10 or 5 classes, respectively.

- **10-Split ImageNet-R** (Hendrycks et al., 2021): ImageNet-R contains 30,000 images from 200 ImageNet classes, covering diverse styles such as art, cartoons, and sketches. It is randomly divided into 10 tasks, each consisting of 20 classes.

- **10-Split DomainNet** (Peng et al., 2019): DomainNet is a cross-domain dataset covering 345 everyday object classes across six domains (clipart, real, sketch, infograph, painting, and quickdraw). Since the number of images per class varies significantly, we follow the exsiting setting (Gao et al., 2024; Lu et al., 2024) by selecting the 200 largest classes and randomly splitting them into 10 tasks, each containing 20 classes with samples drawn from multiple domains.

- **6-Split DomainNet** (Peng et al., 2019): To evaluate robustness under strong distribution shifts, we construct a domain-incremental protocol where each task corresponds to one of the six visual domains in DomainNet (Real, Sketch, Painting, Clipart, Infograph, Quickdraw). For each domain, we select the 20 classes with the largest sample counts to form representative tasks.

## E.2  METRICS

In continual learning, evaluation is typically based on the accuracy matrix $\boldsymbol{A} \in \mathbb{R}^{T \times T}$, where $T$ is the total number of tasks and the entry $A_{i,j}$ denotes the accuracy on task $j$ after training on task $i$. We report two standard metrics: Last Average Accuracy and Last Average Forgetting.

- **Last Average Accuracy (ACC)** measures the overall performance on all tasks after completing the training sequence:

$$\text{ACC} = \frac{1}{T} \sum_{j=1}^{T} A_{T,j},$$

- **Last Average Forgetting (Forgetting)** quantifies the average performance drop on past tasks after learning subsequent ones:

$$\text{Forgetting} = \frac{1}{T-1} \sum_{j=1}^{T-1} \max_{i \in \{1,\dots,T-1\}} (A_{i,j} - A_{T,j}).$$

### E.3 IMPLEMENTATION DETAILS

We follow the class-incremental learning protocol, where task classes are disjoint and the task identity is unknown at inference. Our main experiments are conducted with ViT-B/16 (Dosovitskiy et al., 2020) pretrained on ImageNet-21K as the backbone. In addition, we also report results on two other backbones: ViT-L/16 pretrained with AugReg (Steiner et al., 2021) and the self-supervised DINOv2 ViT-B/14 (Oquab et al., 2023). Each transformer layer is augmented with four learnable prompts. The prompt embeddings are shared across tasks and initialized from a uniform distribution in $[-1, 1]$.

Models are trained with Adam (Kingma & Ba, 2015) (learning rate of 0.01, a batch size of 256, $\beta_1 = 0.9$, $\beta_2 = 0.999$, weight decay $5 \times 10^{-5}$). Each task is trained for 10 epochs on all benchmarks, which is sufficient for convergency (as shown in Appendi E.4). NSP$^2$ is reproduced from its official codebase and hyperparameters, with the sole modification of training for 10 epochs per task instead of the originally reported 100. Empirical evidence in Appendix E.4 confirms fully convergence, disentangling forgetting from underfitting. The training objective combines cross-entropy loss with a feature standard-deviation alignment loss for prompt embeddings (see eq. (28)). The alignment weight is set to $\lambda = 1.0$ for 10-Split CIFAR-100, 20-Split CIFAR-100, 10-Split ImageNet-R, 10-Split DomainNet, and to $\lambda = 2.0$ for 6-Split DomainNet. The temperature parameter in cross-entropy is tuned via cross-validation and set to 28, 25, 30, 30, and 30 for 10-Split CIFAR-100, 20-Split CIFAR-100, 10-Split ImageNet-R, 10-Split DomainNet, and 6-Split DomainNet, respectively.

### E.4 CONVERGENCE GUARANTEE

To justify the training protocol adopted in our experiments, we report the training accuracy and loss curves of both NSP$^2$ and CNSP across tasks on four benchmarks (10-Split-CIFAR100, 20-Split-CIFAR100, 10-Split-DomainNet, and 10-Split-ImageNet-R). As shown in figs. 5 to 8, all curves exhibit rapid convergence within the first 5 epochs: training accuracy quickly increases and stabilizes, while training loss consistently decreases and plateaus. By epoch 10, both methods have fully converged across tasks and datasets, as further confirmed by the shaded regions indicating the last four epochs and the black curve representing the average accuracy/loss across all tasks at each epoch. These results provide empirical evidence that training each task for 10 epochs is sufficient to ensure convergence, without sacrificing performance or stability.

### E.5 ABLATION STUDY

Table 5 reports both the overall comparison and the ablation of CNSP. The lower bound (naïve VPT without continual learning mechanisms) suffers from severe forgetting, while joint training serves as an oracle upper bound. NSP$^2$ provides strong performance, yet CNSP consistently achieves higher ACC and lower forgetting across all four benchmarks, substantially narrowing the gap toward the oracle while clearly surpassing naïve tuning.

The ablation study further clarifies each component's role. Removing the variance preservation loss ($\mathcal{L}_{\text{std}}$) causes moderate ACC drops and increased forgetting, suggesting that it contributes mainly to stability. In contrast, removing the right-side projection ($B_R$) leads to pronounced degradation on all benchmarks, confirming it as the key mechanism for effective feature preservation. Finally, the VPT-only baseline performs worst overall, highlighting the substantial gains achieved by CNSP beyond a frozen backbone.

### E.6 ANALYSIS OF PROMPT EXPRESSIVE CAPACITY

To assess whether the null-space projection restricts the expressive capacity of prompts as the number of tasks increases, we analyze the evolution of the prompt-update subspace over tasks. Specifi-

Table 5: Overall comparison and ablation study of CNSP on four benchmarks by removing the variance preservation loss ($\mathcal{L}_{\text{std}}$), the right-side null-space projection ($\boldsymbol{B_R}$), and the entire CNSP module ("Lower Bound"). Upper Bound indicates training with access to all task data simultaneously.

| Method | 10-Split CIFAR100 | | 20-Split CIFAR100 | | 10-Split ImageNet-R | | 10-Split DomainNet | |
|---|---|---|---|---|---|---|---|---|
| | ACC($\uparrow$) | Forgetting($\downarrow$) | ACC($\uparrow$) | Forgetting($\downarrow$) | ACC($\uparrow$) | Forgetting($\downarrow$) | ACC($\uparrow$) | Forgetting($\downarrow$) |
| Upper Bound | 93.59 | 0.00 | 93.59 | 0.00 | 84.62 | 0.00 | 89.52 | 0.00 |
| NSP[2] | 90.60 | 4.52 | 88.12 | 5.11 | 79.60 | 3.95 | 83.92 | 8.31 |
| **CNSP(Ours)** | 91.10 | 4.43 | 88.76 | 4.79 | 79.75 | 3.93 | 84.49 | 6.92 |
| w/o $\mathcal{L}_{\text{std}}$ | 90.58 | 5.17 | 88.22 | 6.13 | 79.72 | 4.30 | 84.29 | 7.58 |
| w/o $\boldsymbol{B_R}$ | 85.01 | 14.42 | 84.80 | 14.84 | 73.98 | 17.62 | 78.29 | 19.85 |
| Lower Bound | 83.72 | 16.17 | 83.88 | 15.55 | 73.33 | 18.33 | 74.30 | 24.94 |

cally, we examine the rank of the projection matrices $\{\boldsymbol{B}_t^{(l)}\}_{l=1}^{L}$, where $\boldsymbol{B}_t^{(l)}$ denotes the projection matrix used for task $t$ in layer $l$. These matrices determine the allowable update directions at each layer $l$. We define the average rank ratio across layers for each incremental task $t$ ($> 1$) as:

$$\rho_t = \frac{1}{L} \sum_{l=1}^{L} \frac{\text{rank}\left(\boldsymbol{B}_t^{(l)}\right)}{D},$$

where $D$ is the token embedding dimension. A high value of $\rho_t$ indicates that most update directions remain available, implying that the expressive capacity of prompts is preserved.

Table 6 reports the average rank ratio across tasks for three benchmarks using ViT-B/16-21K. Across all setting, $\rho_t$ remains extremely high ($>97.52\%$) and shows no degradation as tasks progress. These results demonstrate that CNSP does not collapse the expressive capacity of prompts. Instead, the null-space constraint removes only the directions that would violate feature preservation, while maintaining a near–full-rank update subspace and strong adaptability throughout continual learning.

Table 6: Average rank ratio (%) of projection matrices across layers for each task.

| Benchmark | $\rho_2$ | $\rho_3$ | $\rho_4$ | $\rho_5$ | $\rho_6$ | $\rho_7$ | $\rho_8$ | $\rho_9$ | $\rho_{10}$ |
|---|---|---|---|---|---|---|---|---|---|
| 10-Split ImageNet-R | 97.53 | 97.53 | 97.53 | 97.54 | 97.53 | 97.53 | 97.54 | 97.53 | 97.53 |
| 10-Split CIFAR100 | 97.53 | 97.54 | 97.54 | 97.54 | 97.54 | 97.54 | 97.54 | 97.55 | 97.55 |
| 10-Split DomainNet | 97.52 | 97.54 | 97.53 | 97.53 | 97.53 | 97.54 | 97.53 | 97.53 | 97.53 |

# F  DISCUSSION AND LIMITATIONS

## F.1  DISCUSSION ON HEAD PRESERVATION

In CNSP, head preservation (eq. (9)) naturally ensured under the standard protocol of using task-specific classification heads in prompt-based continual learning. As illustrated in fig. 4, during training only the head associated with the current task is updated, while all previously learned heads remain frozen. Combined with the feature-preservation condition established in our framework, this implies that every previously trained head receives invariant representations and therefore retains its original input–output mapping, directly satisfying proposition 2.

Beyond the theoretical formulation, the practical cost of employing multiple task-specific heads is negligible. In a ViT-B/16 ($\approx$86M parameters), prompts (four per layer) contribute $\approx$0.037M and a 20-way classifier $\approx$0.015M. Even with ten tasks, the cumulative head parameters remain below 0.2% of the backbone, indicating that the multi-head design does not materially increase model size.

Importantly, CNSP itself does not rely on any particular choice of classifier head. Because our consistency formulation cleanly separates feature preservation from head behavior, the framework is compatible with a wide range of head architectures, including prototype-based, cosine-normalized, low-rank, or other more compact or shared designs. In this work, we adopt the standard independent linear heads primarily to align with common practice in prompt-based continual learning and to ensure a fair comparison with existing baselines, rather than due to any methodological constraint imposed by CNSP.

At inference, we adopt the common practice of concatenating logits from all task-specific heads and selecting the class with the highest confidence. This procedure can be interpreted probabilistically: each head produces calibrated class scores over its own label subset, and concatenation yields a unified score vector over the global label space, upon which a global decision is made via $\arg\max$.

A practical concern is that logits produced by different heads may not be calibrated to the same scale. While this issue is independent of the CNSP consistency theory, it may affect fairness across tasks in the concatenation step. To mitigate this, we apply temperature scaling (Guo et al., 2017) during training to smooth predictions and improve cross-head comparability. More advanced calibration techniques (e.g., Dirichlet calibration (Kull et al., 2019)) or normalization-based heads (e.g., cosine/prototype classifiers) can be readily integrated into CNSP without modifying the underlying consistency formulation.

Finally, when tasks contain visually similar classes, multiple heads may assign high confidence to the same example. Addressing such conflicts—e.g., via logit normalization, lightweight gating, or shared prototype heads—presents an interesting avenue for future work. Importantly, these strategies operate on top of CNSP's theory rather than altering it, enabled by the unified feature and head preservation guarantees established in our framework.

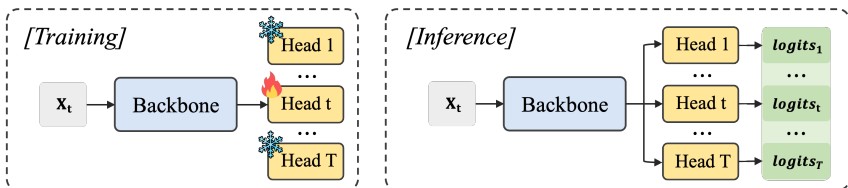

Figure 4: Training and inference of task-specific heads. During training, only the current task head is updated while others remain frozen. During inference, all heads run in parallel and their logits are concatenated, with the final prediction given by the most confident class, eliminating the need for task identity.

## F.2 DISCUSSION ON PROMPT POSITION

In practice, we adopt the strategy of appending prompts after the image patches, following NSP[2] (Lu et al., 2024), to ensure direct comparability. From lemmas 1 and 3, it follows that the preservation conditions in propositions 1 and 2 are invariant to the relative position of prompt tokens within the input sequence. In other words, whether prompts are prepended before the [CLS] token or appended after the patch embeddings, the algebraic form of the preservation constraints remains unchanged. This invariance arises from the row-wise independence of LayerNorm and MLP operations, which treat each token identically regardless of ordering. While these properties hold in theory, a systematic empirical investigation into how different prompt insertion strategies influence performance preservation is beyond the scope of this work. We consider this an interesting and important direction for future research.

## F.3 FUTURE DIRECTIONS AND LIMITATIONS

While CNSP provides a unified theoretical view of feature and head preservation, several aspects remain outside the scope of this work. First, although our analysis gives exact treatments of LayerNorm and MHSA, we keep the softmax-induced constraints analytically tractable by not fully relaxing all nonlinear interactions; consequently, our sufficient conditions remain conservative rather than approaching necessary-and-sufficient characterizations. Second, we adopt task-specific classi-

fication heads for theoretical clarity; scaling to highly heterogeneous tasks may benefit from shared or normalized head designs. Third, our experiments focus on single-modality vision benchmarks.

These scope boundaries point toward several promising extensions. Natural next steps include multi-modal continual learning with vision–language models, adaptive prompt allocation under task heterogeneity, and tighter theoretical analysis of softmax and other nonlinear components within Transformer-based continual learning.

## G    LLM USAGE STATEMENT

We acknowledge the use of Large Language Models (LLMs) as an assistive tool in the preparation of this work. Their role was limited to grammar refinement, LaTeX formatting assistance, language polishing, and minor code debugging/modification. No part of the research methodology, experiments, or data analysis was delegated to LLMs. All conceptual contributions, design of experiments, and interpretation of results were carried out by the authors. The authors take full responsibility for all content presented in this paper, including any text or code that may have been refined with the assistance of LLMs.

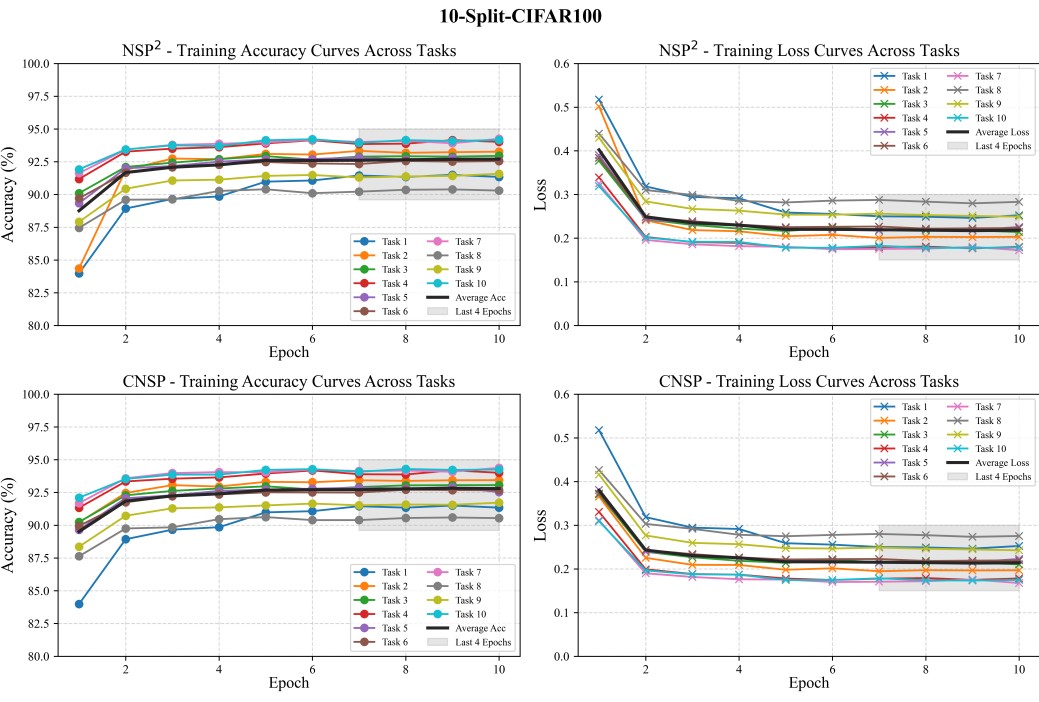

Figure 5: Training accuracy and loss curves across 10 tasks on the 10-Split-CIFAR100 benchmark. The top row reports results of $NSP^2$, and the bottom row reports CNSP. Left: training accuracy across tasks. Right: training loss across tasks. Each colored curve corresponds to one task, while the black curve denotes the average across all tasks at each epoch.

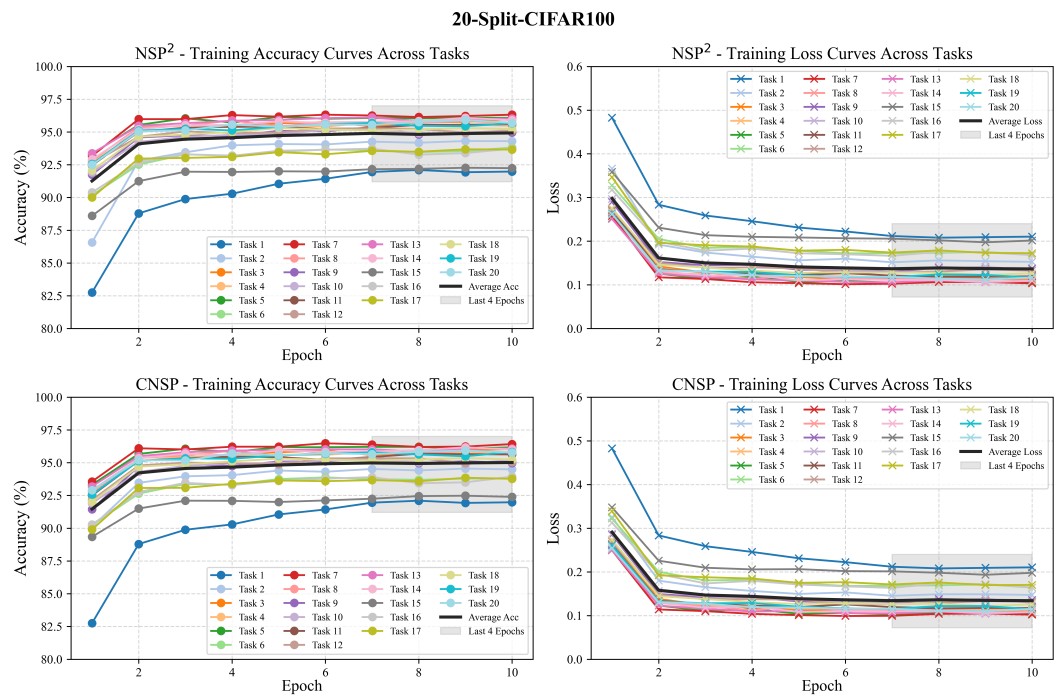

Figure 6: Training accuracy and loss curves across 20 tasks on the 20-Split-CIFAR100 benchmark. The top row reports results of NSP$^2$, and the bottom row reports CNSP. Left: training accuracy across tasks. Right: training loss across tasks. Each colored curve corresponds to one task, while the black curve denotes the average across all tasks at each epoch.

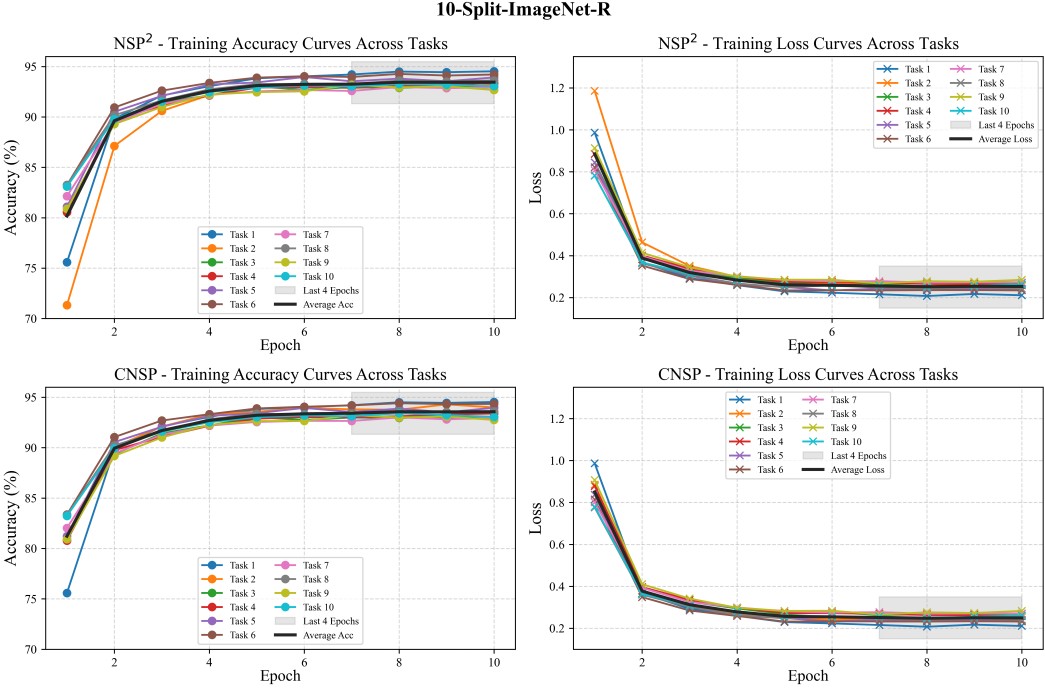

Figure 7: Training accuracy and loss curves across 10 tasks on the 10-Split-ImageNet-R benchmark. The top row reports results of NSP$^2$, and the bottom row reports CNSP. Left: training accuracy across tasks. Right: training loss across tasks. Each colored curve corresponds to one task, while the black curve denotes the average across all tasks at each epoch.

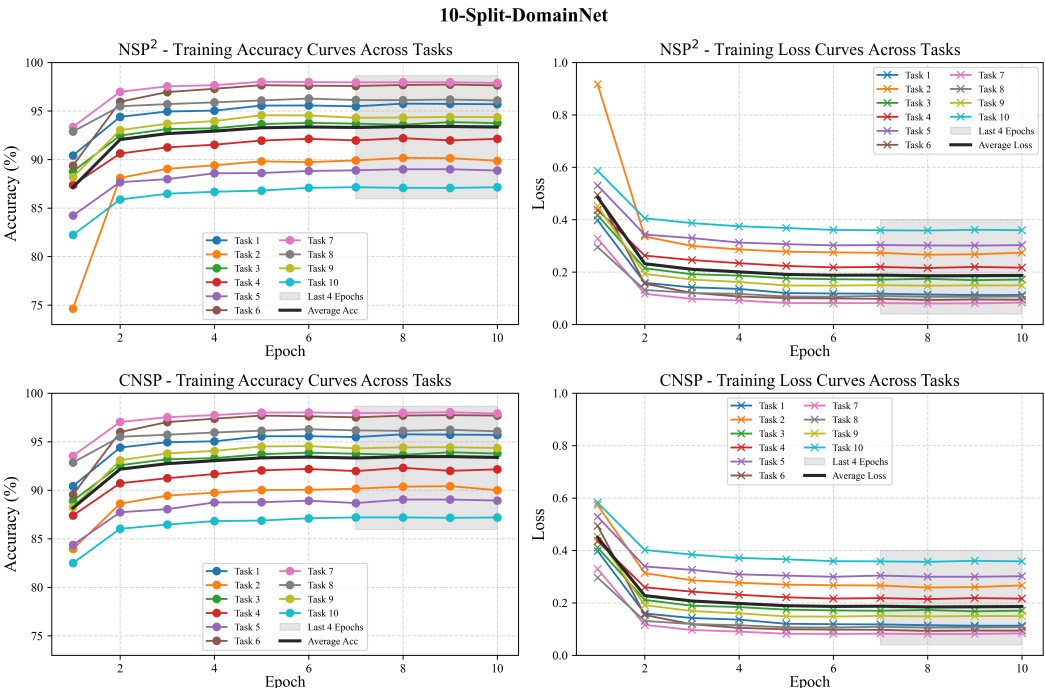

Figure 8: Training accuracy and loss curves across 10 tasks on the 10-Split-DomainNet benchmark. The top row reports results of NSP$^2$, and the bottom row reports CNSP. Left: training accuracy across tasks. Right: training loss across tasks. Each colored curve corresponds to one task, while the black curve denotes the average across all tasks at each epoch.

