# OpenReview forum: "CNSP: Consistent Null-Space Projection for Principled Prompt-Based Continual Learning"
_ICLR.cc/2026/Conference — Submitted to ICLR 2026_

### Official Review · Reviewer_yBKo · 2025-10-19

**Soundness:** 3
**Presentation:** 3
**Contribution:** 2
**Rating:** 4
**Confidence:** 4

**Summary:**

The paper introduces CNSP (Consistent Null-Space Projection), a principled framework for prompt-based continual learning with Vision Transformers. Building on NSP^2, the authors strengthen its theoretical foundations by providing per-head sufficient conditions for multi-head attention, a matrix-form treatment of LayerNorm, and a relaxed variance-only constraint for prompt updates. They also ensure end-to-end consistency by incorporating classification head preservation. Experiments show consistent gains in accuracy and reduced forgetting compared to prior methods, including NSP^2.

**Strengths:**

1. This paper provides an extensive analysis of existing null-space projection method in continual learning (especially NSP^2), and propose corresponding improvements.

2. The proposed method is derived from theoretical basis of feature preservation and head preservation, and seems reasonable.

**Weaknesses:**

1. From my understanding, this work is built on existing null-space projection method NSP^2, which limits its technical novelty.

2. The proposed designs, although very extensive, achieve only marginal improvements over NSP^2.

3. In Table 2 (ablation study), the authors seem to remove major components of null-space projection method. However, as the extension of NSP^2, each individual design should be compared with the counterpart in NSP^2.

4. The authors only consider ViT-B/16 pretrained on ImageNet-21K as the backbone. Does the proposed method also apply to other backbones especially self-supervised checkpoints?

**Questions:**

My major concerns lie in the effectiveness of individual designs over NSP^2. Please refer to the Weaknesses.

---

> ### Author Response · Authors · 2025-11-25
>
> We thank the reviewer for the thoughtful feedback and for highlighting the strengths of our work, particularly the theoretical grounding based on feature and head preservation. We appreciate your constructive suggestions, which have helped us further improve the clarity and completeness of the paper. Below, we address each of your comments in detail:
>
> > Q1. From my understanding, this work is built on existing null-space projection method NSP$^2$, which limits its technical novelty.
>
> We thank the reviewer for the comment. While CNSP is implemented through a projection operator, its goal and theoretical foundation differ substantially from those of NSP². NSP² offers an insightful first attempt at analyzing prompt tuning through null-space conditions, but its derivation treats multi-head attention and LayerNorm under simplified assumptions and excludes the classification head from the overall consistency formulation.
>
> CNSP revisits this problem from an end-to-end, matrix-form perspective and provides the first unified and mathematically complete framework for task-performance preservation in prompted Transformers. Specifically, CNSP derives consistency conditions jointly for MHSA, LayerNorm, and the classification head under full Transformer parameterization, yielding a principled null-space projection as a direct analytical consequence. This level of theoretical closure is not present in NSP² and reflects a different objective rather than an incremental extension.
>
> We have revised the manuscript to emphasize this distinction more clearly.
>
> > Q2. The proposed designs, although very extensive, achieve only marginal improvements over NSP$^2$.
>
> We appreciate the reviewer’s observation. While some ACC gains appear modest, continual-learning quality is jointly determined by accuracy and forgetting. Under our end-to-end theoretical framework, CNSP achieves consistent improvements in both metrics across all submitted benchmarks, indicating that the derived consistency constraints function as intended.
>
> Importantly, the distinction between CNSP and NSP² becomes clearer in more challenging regimes—settings where continual-learning stability is more difficult to maintain and approximation errors accumulate. In these cases, the benefits of CNSP become more pronounced:
>
> - **Model scaling (ViT-L/16, Table I)**
>   As model dimensionality increases, approximation artifacts become more impactful. For instance, on 10-Split ImageNet-R, CNSP’s exact projection maintains stability, reducing forgetting from 6.50% → 4.29% (34% relative reduction), together with an ACC improvement (83.52% → 84.78%).
>
> - **Severe cross-domain shift (6-Split DomainNet, Table II)**
>   Each task corresponds to a distinct visual domain, leading to substantial distributional change. CNSP maintains consistent robustness, reducing forgetting from 6.67% → 5.84% (12% relative reduction) with a concurrent ACC gain (83.55% → 84.32%).
>
> Taken together, these results show that CNSP delivers steady improvements under standard settings and increasingly larger gains exactly when the continual-learning problem becomes more demanding—precisely the scenarios where rigorous, theoretically grounded consistency constraints are most valuable. We have clarified this perspective and highlighted the additional experiments in the revised manuscript.
>
> **Table I.** Results on four benchmarks with ViT-L/16 (AugReg, IN-21k) backbone.
>
> |     | 10-Split ImageNet-R | 10-Split ImageNet-R | 10-Split DomainNet | 10-Split DomainNet | 10-Split CIFAR100 | 10-Split CIFAR100 | 20-Split CIFAR100 | 20-Split CIFAR100 |
> |:--- | :---: | :---: | :---: |:---:|:---:|:---:|:---:|:---:|
> | **Method** | ACC(↑) | Forgetting(↓) | ACC(↑) | Forgetting(↓) | ACC(↑) | Forgetting(↓) | ACC(↑) | Forgetting(↓) |
> | Upper Bound | 88.58 | 0.00 | 90.48 | 0.00 | 95.06 | 0.00 | 95.06 | 0.00 |
> | Lower Bound | 79.72 | 15.51 | 80.85 | 15.88 | 88.91 | 10.11 | 87.40 | 11.32 |
> | NSP² | 83.52 | 6.50 | 84.38 | 10.28 | 92.20 | 4.14 | 90.83 | 6.34 |
> | **CNSP** | 84.78 | 4.29 | 85.63 | 8.03 | 92.45 | 4.01 | 91.02 | 5.24 |
>
> **Table II.** Experimental results on the 6-Split DomainNet benchmark across three runs with ViT-B/16-21K backbone.
>
> |        | run1 | run1| run2 | run2 | run3 |run3|
> | :--- | :--- | :--- | :--- | :--- | :--- | :--- |
> | **Method** | ACC(↑) | Forgetting(↓) | ACC(↑) | Forgetting(↓) | ACC(↑) | Forgetting(↓) |
> | Upper Bound | 91.68 | 0.00 | 91.94 | 0.00 | 91.88 | 0.00 |
> | Lower Bound | 63.34 | 34.11 | 79.17 | 18.52 | 78.22 | 17.65 |
> | NSP² | 83.55 | 6.67 | 84.85 | 7.91 | 84.76 | 3.36 |
> | **CNSP** | 84.32 | 5.84 | 85.34 | 6.44 | 85.18 | 3.07 |

---

> > ### Author Response · Authors · 2025-11-25
> >
> > > Q3. In Table 2 (ablation study), the authors seem to remove major components of null-space projection method. However, as the extension of NSP$^2$, each individual design should be compared with the counterpart in NSP$^2$.
> >
> > We thank the reviewer for raising this point. We would like to clarify that CNSP is not an extension of NSP², and its components are not designed to correspond one-to-one with those in NSP². Although both methods ultimately employ projection operators, the underlying theoretical bases differ fundamentally. NSP² constructs two projections per layer (a left $M\times M$ and a right $D\times D$ projection) derived from a simplified formulation. By contrast, CNSP begins with a unified end-to-end consistency analysis and derives rigorous sufficient conditions under the full Transformer parameterization. These conditions naturally lead to a single right-side $D\times D$ projection per layer as the analytical consequence of the theory.
> >
> > Because the theoretical assumptions, derivation paths, and resulting projection structures differ conceptually and mathematically, a component-wise pairing between CNSP and NSP² is not well-defined. Our ablation therefore focuses on assessing the empirical contribution of CNSP’s own theoretically motivated components, rather than creating artificial correspondences across methods with different formulations.
> > > Q4. The authors only consider ViT-B/16 pretrained on ImageNet-21K as the backbone. Does the proposed method also apply to other backbones especially self-supervised checkpoints?
> >
> > We thank the reviewer for raising this important question. CNSP is derived purely from the mathematical structure of transformer-based prompt tuning, and therefore applies to any promptable Transformer architecture and any pretraining paradigm, including self-supervised ones. The framework makes no assumptions specific to ImageNet-21K pretraining or to the ViT-B/16 architecture.
> >
> > To empirically validate this generality, we additionally evaluate CNSP on a self-supervised model, DINOv2 ViT-B/14, which differs substantially from supervised ViT-B/16 in both architecture (patch size and embedding dimensionality) and pretraining methodology (fully self-supervised). As shown in Table III, CNSP again consistently improves over NSP² in both ACC and forgetting, confirming that our method generalizes to self-supervised backbones. We have included this experiment in the revised manuscript and made the generality of the framework explicit.
> >
> > **Table III.** Results on four benchmarks with DINOv2 ViT-B/14 backbone.
> >
> > |     | 10-Split ImageNet-R | 10-Split ImageNet-R | 10-Split DomainNet | 10-Split DomainNet | 10-Split CIFAR100 | 10-Split CIFAR100 | 20-Split CIFAR100 | 20-Split CIFAR100 |
> > | --- | --- | --- | --- | --- | --- | --- | --- | --- |
> > | **Method** | ACC(↑) | Forgetting(↓) | ACC(↑) | Forgetting(↓) | ACC(↑) | Forgetting(↓) | ACC(↑) | Forgetting(↓) |
> > | Upper Bound | 89.58 | 0.00 | 89.65 | 0.00 | 93.22 | 0.00 | 93.22 | 0.00 |
> > | Lower Bound | 78.42 | 17.66 | 68.87 | 31.89 | 78.01 | 22.27 | 77.84 | 21.80 |
> > | NSP² | 85.22 | 4.91 | 84.52 | 10.06 | 89.43 | 5.08 | 87.15 | 5.92 |
> > | **CNSP** | 85.67 | 4.49 | 85.27 | 8.50 | 89.87 | 4.79 | 87.29 | 5.72 |
> >
> > **Conclusion.**
> > We thank the reviewer for the thoughtful feedback. We have clarified the theoretical motivation, strengthened the explanations, and expanded the empirical analysis accordingly. We hope these revisions help address the concerns raised, and we sincerely appreciate the time and effort dedicated to reviewing our work.

---

> > > ### Comment · Reviewer_yBKo · 2025-11-25
> > >
> > > I thank the authors for their rebuttal. Although it's good to see better forgetting alleviation, I'm not convinced that it's a significant progress given the marginal improvement on average accuracy with the additional complex designs . For continual learning, the final goal is better average accuracy of all seen tasks. The forgetting metric is only part of the problem and is balanced with learning plasticity. Also, even the improvement in forgetting metric is not consistent, e.g., many results on DINOv2 ViT-B/14 backbone are still marginal. I therefore keep my original rating.

---

### Official Review · Reviewer_pGHs · 2025-10-29

**Soundness:** 2
**Presentation:** 3
**Contribution:** 2
**Rating:** 4
**Confidence:** 5

**Summary:**

This paper studies the problem of prompt-based continual learning. It analyzes the limitations of a regularization-based method, NSP2, and identifies several key weaknesses, including the lack of rigorous justification for its extension to multi-head attention, the oversimplified treatment of LayerNorm, the unstable invariance assumption on prompts, and the neglect of classification head analysis. To overcome these issues, the paper proposes Consistent Null-Space Projection, which introduces rigorous multi-head analysis, matrix-form LayerNorm modeling, a relaxed distributional constraint, and classification-head preservation to ensure theoretical consistency. Experimental results show that the proposed method consistently outperforms NSP2 in terms of both average accuracy and forgetting across multiple benchmarks.

**Strengths:**

1. This paper introduces the concepts of feature preservation and head preservation through theoretical analysis, identifying the key factors that maintain the stability of prompt-based continual learning models.

2. In the methodology section, a more rigorous derivation of multi-head attention is presented, leading to a set of sufficient conditions for feature preservation in VPT.

3. The analysis reformulates LayerNorm’s broadcast operations in matrix form, thereby ensuring greater algebraic rigor and theoretical soundness.

4. CNSP achieves performance competitive with state-of-the-art prompt-based continual learning methods, demonstrating both theoretical and empirical advantages.

**Weaknesses:**

1. Since each attention head operates independently, the per-head analysis conducted in this paper essentially follows the same procedure as in NSP2, except that NSP2 did not explicitly denote each head with the superscript (h). As a result, the derivation ultimately reduces to single-head-level results rather than a unified multi-head formulation. This head-level analysis does not provide substantial new insights.

2. The adoption of a right-side nullification form on the second constraint appears questionable. The purpose of the second condition in Eq. (27) is to ensure that the attention output (i.e., the product of the attention matrix and the value matrix V) remains invariant. However, by removing the attention matrix S from the formulation, this invariance is no longer guaranteed, weakening the theoretical justification for the constraint.

3. The claim of “classification head preservation by design” lacks meaningful novelty. This setup does not require any special design, as most continual learning methods already adopt the same practice (i.e., training task-specific heads and concatenating them at inference).

4. The experimental results are limited, consisting mainly of a single comparison with the current state-of-the-art and one ablation study (in the appendix). Moreover, as shown in Table 1, the improvements over NSP2 are marginal and not particularly significant.

**Questions:**

1. In Eq. (23), both gamma and beta are defined as element-wise parameters. However, in Eq. (24), after reformulating them into matrix form, they are expressed as if they participate in matrix multiplication. This seems inconsistent with their element-wise nature. The paper should provide a clear explanation for this formulation choice, as it could directly affect the validity of the subsequent derivations and conclusions.

2. Eq. (25) introduces the variance-invariance assumption, but in transitioning from Eq. (23) to (24), the mean term $\mu_{P_t+\Delta P}$ is omitted. It is unclear whether this omission is mathematically equivalent to the original formulation (Eq. 23). If not, then the assumption in Eq. (25), which constrains only the variance while ignoring the mean, may not be strictly valid, potentially weakening the theoretical rigor of the analysis.

---

> ### Author Response · Authors · 2025-11-25
>
> We thank the reviewer for the thoughtful and constructive feedback. We appreciate the recognition of the theoretical formulation—particularly the roles of feature and head preservation, the strengthened derivations for MHSA and LayerNorm—and the competitive performance of CNSP. We address each of your comments in detail below.
>
> > Q1. Since each attention head operates independently, the per-head analysis conducted in this paper essentially follows the same procedure as in NSP2, except that NSP2 did not explicitly denote each head with the superscript (h). As a result, the derivation ultimately reduces to single-head-level results rather than a unified multi-head formulation. This head-level analysis does not provide substantial new insights.
>
> We thank the reviewer for raising this point. Although the final sufficient condition is expressed per head, a full multi-head analysis is essential for establishing it rigorously under the MHSA composition—especially the concatenation structure across heads and the shared output projection $\mathbf{W}_o$. These interactions couple the heads in ways that cannot be justified through single-head reasoning alone. NSP² does not model this multi-head composition explicitly. In contrast, CNSP introduces a principled multi-head formulation that:
> - Derives when per-head constraints are sufficient for preserving the MHSA output under prompt updates;
> - Characterizes the full admissible constraint space, showing how the head-wise condition arises as a tractable and theoretically grounded subset; and
> - Invites further investigation into potential stronger joint-head constraints, providing a foundation for future extensions.
>
> Thus, the contribution of our MHSA treatment is not merely the per-head condition itself, but the theoretical framework that justifies its validity and delineates its scope within the full multi-head architecture. We have updated the revised manuscript to make this point clearer.
>
> > Q2. The adoption of a right-side nullification form on the second constraint appears questionable. The purpose of the second condition in Eq. (27) is to ensure that the attention output (i.e., the product of the attention matrix and the value matrix V) remains invariant. However, by removing the attention matrix S from the formulation, this invariance is no longer guaranteed, weakening the theoretical justification for the constraint.
>
> We thank the reviewer for raising this point.
> In Sec. 4.3, we clarify the motivation for adopting the right-side nullification form when implementing the second constraint in Eq. (27). The key reason is to avoid the circular dependence introduced by the LayerNorm variance term $\mathbf{\sigma}$, which couples the prompt update with the very statistics $\mathbf{\sigma}$ that must be aligned through the standard-deviation alignment loss.
>
> Importantly, this implementation does not weaken the theoretical justification. The right-side constraint $\Delta \mathbf{P}\mathbf{C}\mathbf{\Gamma}\mathbf{W}_v^{(h)}=0$ remains a valid sufficient condition for the second term in Eq. (27). Specifically,
> $\Delta \mathbf{P} \mathbf{C} \mathbf{\Gamma} \mathbf{W}_v ^{(h)}=0 \Rightarrow  \mathbf{S} _{\mathbf{X} _t\mathbf{P} _t} ^{(h)} \mathbf{D} _{\mathbf{\sigma}} ^{(-1)} \Delta \mathbf{P}\mathbf{C}\mathbf{\Gamma}\mathbf{W} _v ^{(h)}=0$,
> which is exactly the invariance required by Eq. (27). Thus, the chosen formulation is a practical instantiation of the theoretical sufficient condition—not a relaxation of it—and it preserves the required attention-output consistency. While alternative instantiations are possible, the adopted form is mathematically sound and empirically stable, as demonstrated by our experimental results.

---

> > ### Author Response · Authors · 2025-11-25
> >
> > > Q3. The claim of “classification head preservation by design” lacks meaningful novelty. This setup does not require any special design, as most continual learning methods already adopt the same practice (i.e., training task-specific heads and concatenating them at inference).
> >
> > We thank the reviewer for the comment. We agree that using task-specific classification heads is a common and effective design in prompt-based continual learning (e.g., L2P, DualPrompt). CNSP does not aim to introduce novelty in this architectural choice.
> > Rather, our contribution lies in providing the theoretical justification for when and why this design is valid within prompt-based continual learning.
> >
> > While prior works employ frozen task-specific heads as a practical convention, the underlying conditions that make this strategy sound—particularly in the presence of MHSA and LayerNorm—have not been made explicit.
> > CNSP formalizes this connection by showing that:
> > - task-performance preservation mathematically decomposes into feature preservation and head preservation,
> > - prompt updates should keep the representations seen by earlier heads invariant, and
> > - under these conditions, freezing previously trained heads and concatenating their logits yields a coherent global decision rule.
> >
> > Thus, the role of the classification head in our framework is not a new architectural component, but a theoretically grounded part of the end-to-end consistency conditions derived in CNSP. This analysis also naturally supports alternative head designs (e.g., cosine or prototype heads), which follow directly from the same theoretical framework. We have clarified this distinction more explicitly in the revised manuscript.
> >
> > > Q4. The experimental results are limited, consisting mainly of a single comparison with the current state-of-the-art and one ablation study (in the appendix). Moreover, as shown in Table 1, the improvements over NSP2 are marginal and not particularly significant.
> >
> > We thank the reviewer for the helpful feedback. To evaluate CNSP under diverse continual-learning conditions, our original submission reported results on four ViT-B/16 benchmarks—10-Split CIFAR100, 20-Split CIFAR100, 10-Split ImageNet-R, and 10-Split DomainNet—covering a wide range of dataset scales, difficulty levels, and domain shifts. While some ACC gains appear moderate, continual-learning performance depends jointly on accuracy and forgetting. Under CNSP’s end-to-end theoretical formulation, we observe consistent improvements in both metrics across all four benchmarks, indicating that the derived constraints behave as intended.
> >
> > To further strengthen the empirical validation, we additionally evaluated CNSP on:
> > - ViT-L/16 (AugReg, IN-21K) to assess model scaling, and
> > - 6-Split DomainNet (one distinct visual domain per task) to assess severe domain heterogeneity.
> >
> > Across these settings (Tables I and II), CNSP again improves over NSP², and the gains become more pronounced in challenging regimes. For example, on ViT-L/16, forgetting decreases from 6.50% → 4.29% on 10-Split ImageNet-R (a 34% relative reduction) with a concurrent ACC improvement (83.52% → 84.78%). On 6-Split DomainNet, forgetting reduces from 6.67% → 5.84% with an ACC gain (83.55% → 84.32%). These patterns indicate that CNSP’s theoretically grounded consistency constraints become more impactful as model dimensionality grows or domain shifts intensify—settings in which approximate treatments tend to accumulate error and stability becomes more critical.
> >
> > Together with the original ViT-B/16 benchmarks, these extended experiments provide a broad and comprehensive evaluation of CNSP. We have integrated and highlighted these results in the revised manuscript.
> >
> > **Table I.** Results on four benchmarks with ViT-L/16 (AugReg, IN-21k) backbone.
> >
> > |     | 10-Split ImageNet-R | 10-Split ImageNet-R | 10-Split DomainNet | 10-Split DomainNet | 10-Split CIFAR100 | 10-Split CIFAR100 | 20-Split CIFAR100 | 20-Split CIFAR100 |
> > |:--- | :---: | :---: | :---: |:---:|:---:|:---:|:---:|:---:|
> > | **Method** | ACC(↑) | Forgetting(↓) | ACC(↑) | Forgetting(↓) | ACC(↑) | Forgetting(↓) | ACC(↑) | Forgetting(↓) |
> > | Upper Bound | 88.58 | 0.00 | 90.48 | 0.00 | 95.06 | 0.00 | 95.06 | 0.00 |
> > | Lower Bound | 79.72 | 15.51 | 80.85 | 15.88 | 88.91 | 10.11 | 87.40 | 11.32 |
> > | NSP² | 83.52 | 6.50 | 84.38 | 10.28 | 92.20 | 4.14 | 90.83 | 6.34 |
> > | **CNSP** | 84.78 | 4.29 | 85.63 | 8.03 | 92.45 | 4.01 | 91.02 | 5.24 |
> >
> > **Table II.** Experimental results on the 6-Split DomainNet benchmark across three runs with ViT-B/16-21K backbone.
> >
> > |        | run1 | run1| run2 | run2 | run3 |run3|
> > | :--- | :--- | :--- | :--- | :--- | :--- | :--- |
> > | **Method** | ACC(↑) | Forgetting(↓) | ACC(↑) | Forgetting(↓) | ACC(↑) | Forgetting(↓) |
> > | Upper Bound | 91.68 | 0.00 | 91.94 | 0.00 | 91.88 | 0.00 |
> > | Lower Bound | 63.34 | 34.11 | 79.17 | 18.52 | 78.22 | 17.65 |
> > | NSP² | 83.55 | 6.67 | 84.85 | 7.91 | 84.76 | 3.36 |
> > | **CNSP** | 84.32 | 5.84 | 85.34 | 6.44 | 85.18 | 3.07 |

---

> > > ### Author Response · Authors · 2025-11-25
> > >
> > > > Q5. In Eq. (23), both gamma and beta are defined as element-wise parameters. However, in Eq. (24), after reformulating them into matrix form, they are expressed as if they participate in matrix multiplication. This seems inconsistent with their element-wise nature. The paper should provide a clear explanation for this formulation choice, as it could directly affect the validity of the subsequent derivations and conclusions.
> > >
> > > We thank the reviewer for this question.
> > > LayerNorm's affine parameters $\mathbf{\gamma}$, $\mathbf{\beta}\in\mathbb{R} ^{D}$ are indeed defined element-wise along the feature dimension $D$ and are shared (broadcast) across the batch dimension. For an input $\mathbf{X}\in \mathbb{R} ^{N\times D}$, the broadcasted affine operation
> > >
> > > $(\mathbf{X}\cdot\mathbf{\gamma})[i,j]=\mathbf{X}[i,j]\mathbf{\gamma}[j]$
> > >
> > > is exactly equivalent to right-multiplication with the diagonal matrix $\mathbf{\Gamma}=\mathrm{diag}(\mathbf{\gamma})$:
> > >
> > > $(\mathbf{X}\mathbf{\Gamma})[i,j] =\sum\limits _{k} \mathbf{X}[i,k] \mathbf{\Gamma}[k,j]=\mathbf{X}[i,j]\mathbf{\gamma}[j]$.
> > >
> > > Thus, the matrix form in Eq. (24) is a strictly equivalent linear-algebraic representation of the standard LayerNorm affine step—it neither changes the operation nor introduces any additional assumptions. This representation preserves the original element-wise semantics while enabling cleaner and more transparent derivations. All subsequent derivations rely only on $\mathbf{\Gamma}$ being a diagonal linear operator, fully consistent with the original element-wise semantics along the feature dimension.
> > >
> > > > Q6. Eq. (25) introduces the variance-invariance assumption, but in transitioning from Eq. (23) to (24), the mean term is omitted. It is unclear whether this omission is mathematically equivalent to the original formulation (Eq. 23). If not, then the assumption in Eq. (25), which constrains only the variance while ignoring the mean, may not be strictly valid, potentially weakening the theoretical rigor of the analysis.
> > >
> > > We thank the reviewer for raising this question. The transition from Eq. (23) to Eq. (24) does not omit the mean term nor introduce any approximation. The mean subtraction (through broadcasting) in LayerNorm is exactly represented by the centering matrix $\mathbf{C}=\mathbf{I} _D-\frac{1}{D}\mathbf{1} _D\mathbf{1} _D ^{\top} $. For any $\mathbf{X}\in\mathbb{R} ^{N\times D}$ with row-wise means $\mathbf{\mu} _{\mathbf{X}}\in\mathbb{R} ^{N}$, we have:
> > >
> > > $\mathbf{X}\mathbf{C}=\mathbf{X}\left(\mathbf{I} _D-\frac{1}{D}\mathbf{1} _D\mathbf{1} _D^{\top}\right)=\mathbf{X}-\frac{1}{D}\mathbf{X}\mathbf{1} _D\mathbf{1} _D ^{\top}$,
> > >
> > > and element-wise:
> > >
> > > $(\mathbf{X} \mathbf{C})[i,j]=\mathbf{X}[i,j]-\frac{1}{D} \sum\limits _{k=1} ^D\mathbf{X}[i,k]=\mathbf{X}[i,j]-\mathbf{\mu} _{\mathbf{X}}[i]$,
> > >
> > > which is exactly the broadcasted mean subtraction in standard LayerNorm.
> > >
> > > Thus, the centering step in Eq. (23) is exactly preserved in Eq. (24); the mean term is not dropped but absorbed into the linear operator $\mathbf{C}$. The reformulation is therefore mathematically equivalent while enabling a more compact and transparent derivation.
> > >
> > > **Conclusion.** We thank the reviewer for the thoughtful comments. Your feedback has helped improve the clarity of the paper, and we sincerely appreciate the time and care taken in reviewing our work.

---

> ### Comment · Reviewer_pGHs · 2025-11-26
>
> Thank you very much for the response. I have read the rebuttal coments, I lower down my score mainly interms of the marginal improvements of the experimental results, and the incremental novelty which is just detailed extesion of the baseline method.
> This paper seems like something made up and forced. The actual contribution isn't that significant.

---

### Official Review · Reviewer_3yso · 2025-10-30

**Soundness:** 3
**Presentation:** 3
**Contribution:** 2
**Rating:** 4
**Confidence:** 3

**Summary:**

This paper introduces Consistent Null-Space Projection (CNSP), a novel prompt-based continual learning method. Building upon NSP2, the method systematically analyzes multi-head attention and LayerNorm, thereby enhancing the theoretical foundation for knowledge retention through null-space projection. Extensive experimental evaluations demonstrate that CNSP achieves state-of-the-art performance across multiple benchmark datasets.

**Strengths:**

1.	The motivation behind this paper is both intuitive and clearly articulated. In contrast to previous works, CNSP provides a thorough investigation of multi-head attention, LayerNorm, and the classification head within the context of continual learning. Specifically, the paper introduces a more principled and effective framework for visual prompt tuning with vision transformers.
2.	This paper includes theoretical derivations and proofs.
3.	The paper is written in a clear and well-organized manner.

**Weaknesses:**

1.	Compared to NSP2, the paper provides a more systematic theoretical analysis of multi-head attention, LayerNorm, and the classification head. However, the experimental results show only modest improvements.
2.	The contribution appears to be somewhat limited and does not fully meet the conference's expectations. The method seems to be an instantiation of orthogonal projection techniques, applied to continual learning in image classification, and is based on visual prompt tuning with vision transformers.
3.	Maintaining a separate classification head for each task leads to an increase in the number of parameters as the number of tasks grows. Furthermore, calibrating the logits obtained from independently trained classification heads across different tasks presents a new challenge.
4.	There is a notation error in lines 248 and 654. The correct index should be A(2) instead of A(1).

**Questions:**

1.	Please clarify the core contribution of the work, rather than describing the differences in unexplored components from previous works, such as multi-head attention, LayerNorm, and the classification head.
2.	From the ablation study in Table 2, the variance preservation loss has minimal impact on the results. Does this suggest that the assumptions in Equation 25 do not significantly affect the derivation of the final constraints?

---

> ### Author Response · Authors · 2025-11-25
>
> We thank the reviewer for the thoughtful and constructive feedback, and we are glad that the motivation, theoretical rigor, and organization of the paper were well received. We address your points in detail below.
>
> > Q1. Compared to NSP$^2$, the paper provides a more systematic theoretical analysis of multi-head attention, LayerNorm, and the classification head. However, the experimental results show only modest improvements.
>
> We thank the reviewer for the constructive feedback.
> CNSP yields consistent improvements in both ACC and forgetting across all benchmarks, indicating that the derived consistency constraints function as intended. These uniform gains—obtained even on strong pretrained backbones and without any architectural expansion—demonstrate that the theoretical conditions translate effectively into stable and beneficial prompt updates in practice.
>
> At the same time, the extended experiments (Tables I and II) reveal scenarios where the benefits of CNSP become more pronounced. On ViT-L/16 (AugReg, IN-21K), which introduces substantially higher model dimensionality, and on 6-Split DomainNet, where each task corresponds to a distinct visual domain (Real, Sketch, Clipart, Infograph, Painting, Quickdraw), approximate treatments tend to accumulate error and stability becomes more challenging. For instance, on ViT-L/16, forgetting decreases from 6.50% → 4.29% on 10-Split ImageNet-R (a 34% relative reduction) with a concurrent ACC improvement (83.52% → 84.78%). On 6-Split DomainNet, forgetting reduces from 6.67% → 5.84% with an ACC gain (83.55% → 84.32%). In these settings, CNSP shows clearer improvements over NSP², highlighting the advantages offered by its end-to-end consistency constraints as task difficulty or model expressiveness increases.
>
> Across all evaluated backbones and benchmarks, CNSP yields consistent improvements in both accuracy and forgetting, demonstrating that the theoretically derived constraints are effective without requiring additional heuristics or task-specific tuning. These results are now included and highlighted in the revised manuscript.
>
> **Table I.** Results on four benchmarks with ViT-L/16 (AugReg, IN-21k) backbone.
>
> |     | 10-Split ImageNet-R | 10-Split ImageNet-R | 10-Split DomainNet | 10-Split DomainNet | 10-Split CIFAR100 | 10-Split CIFAR100 | 20-Split CIFAR100 | 20-Split CIFAR100 |
> |:--- | :---: | :---: | :---: |:---:|:---:|:---:|:---:|:---:|
> | **Method** | ACC(↑) | Forgetting(↓) | ACC(↑) | Forgetting(↓) | ACC(↑) | Forgetting(↓) | ACC(↑) | Forgetting(↓) |
> | Upper Bound | 88.58 | 0.00 | 90.48 | 0.00 | 95.06 | 0.00 | 95.06 | 0.00 |
> | Lower Bound | 79.72 | 15.51 | 80.85 | 15.88 | 88.91 | 10.11 | 87.40 | 11.32 |
> | NSP² | 83.52 | 6.50 | 84.38 | 10.28 | 92.20 | 4.14 | 90.83 | 6.34 |
> | **CNSP** | 84.78 | 4.29 | 85.63 | 8.03 | 92.45 | 4.01 | 91.02 | 5.24 |
>
> **Table II.** Experimental results on the 6-Split DomainNet benchmark across three runs with ViT-B/16-21K backbone.
>
> |        | run1 | run1| run2 | run2 | run3 |run3|
> | :--- | :--- | :--- | :--- | :--- | :--- | :--- |
> | **Method** | ACC(↑) | Forgetting(↓) | ACC(↑) | Forgetting(↓) | ACC(↑) | Forgetting(↓) |
> | Upper Bound | 91.68 | 0.00 | 91.94 | 0.00 | 91.88 | 0.00 |
> | Lower Bound | 63.34 | 34.11 | 79.17 | 18.52 | 78.22 | 17.65 |
> | NSP² | 83.55 | 6.67 | 84.85 | 7.91 | 84.76 | 3.36 |
> | **CNSP** | 84.32 | 5.84 | 85.34 | 6.44 | 85.18 | 3.07 |
>
> > Q2. The contribution appears to be somewhat limited and does not fully meet the conference's expectations. The method seems to be an instantiation of orthogonal projection techniques, applied to continual learning in image classification, and is based on visual prompt tuning with vision transformers.
>
> Thank you for the helpful comment. It gives us a chance to clarify that CNSP’s contribution lies in its unified theoretical framework, from which the projection form naturally follows. While CNSP employs a projection operator in its final implementation, the contribution of the method is not the projection step itself. It arises directly as the analytical consequence of rigorously deriving task-wise consistency conditions under full Transformer parameterization.
>
> Accordingly, the central contribution of CNSP is the theoretical framework itself: it offers the first mathematically coherent formulation showing that these components jointly admit a unified null-space consistency condition for prompt-based continual learning. Moreover, CNSP’s formulation is architectural rather than task-specific. Because it constrains prompt-induced representational drift through the general MHSA + LayerNorm structure shared across Transformers, the same theoretical reasoning chain applies directly to transformer-based architectures used for segmentation, detection, and other tasks. We have highlighted these points more clearly in the revised version.

---

> > ### Author Response · Authors · 2025-11-25
> >
> > > Q3. Maintaining a separate classification head for each task leads to an increase in the number of parameters as the number of tasks grows. Furthermore, calibrating the logits obtained from independently trained classification heads across different tasks presents a new challenge.
> >
> > We thank the reviewer for raising this point.
> >
> > Task-specific classification heads are a standard design choice in prompt-based continual learning (e.g., L2P, DualPrompt), and we follow this protocol to ensure a fair comparison. The additional parameters introduced by these heads are negligible relative to the transformer backbone. For ViT-B/16, even with 10 tasks, head parameters remain <0.16M (<0.2% of the 86M backbone) and thus do not affect scalability.
> >
> > More importantly, head design is orthogonal to the core contribution of CNSP. Our framework focuses on representational stability and provides a rigorous, end-to-end analysis of how MHSA, LayerNorm, and the classification head jointly constrain permissible prompt updates. Under this formulation, CNSP formally integrates head preservation into the global consistency condition, offering the principled explanation for when independently trained task-specific heads remain valid—an aspect not theoretically addressed in prior prompt-based methods.
> >
> > This theoretical grounding also means that logit calibration is decoupled from CNSP: because features for old heads are provably unchanged, any calibration or normalization mechanism (e.g., cosine-normalized heads, energy-based scaling) can be applied independently if desired. Likewise, alternative classifier designs—such as compressed heads, shared-prototype heads, projection-based heads, or prompting-based heads—can be incorporated without modifying the core consistency formulation, representing promising directions for future extensions enabled by CNSP.
> >
> > > Q4. There is a notation error in lines 248 and 654. The correct index should be A(2) instead of A(1).
> >
> > We thank the reviewer for catching this notation error. We have corrected it in the revised manuscript.
> >
> > > Q5. Please clarify the core contribution of the work, rather than describing the differences in unexplored components from previous works, such as multi-head attention, LayerNorm, and the classification head.
> >
> > We thank the reviewer for giving us the opportunity to clarify the core contribution of our work.
> >
> > The main contribution of CNSP is not the individual analyses of MHSA, LayerNorm, or the classification head, but the unified theoretical framework that connects them. CNSP is the first approach to show that prompt-based continual learning in Transformers is governed by a single end-to-end representational consistency principle, from which the projection rule follows naturally.
> >
> > Concretely:
> >
> > **(1) Unified theoretical closure.**
> > CNSP establishes the first mathematically coherent formulation showing that task-performance preservation decomposes into feature preservation + head preservation, providing end-to-end consistency across the entire Transformer.
> >
> > **(2) Sufficient consistency conditions.** By rigorously modeling MHSA, LayerNorm, and classifier behavior using an algebraically precise matrix-form characterization under full prompted Transformer parameterization, we derive explicit sufficient conditions for past-task performance preservation, leading to a tractable null-space projection rule.
> >
> > **(3) Consistent empirical gains.** CNSP reliably reduces forgetting and improves accuracy across diverse benchmarks and backbones, with more noticeable gains in high-dimensional and cross-domain settings—all without introducing architectural heuristics or task-specific prompt designs.
> >
> > We have revised the manuscript to state this overarching contribution more explicitly.

---

> > > ### Author Response · Authors · 2025-11-25
> > >
> > > > Q6. From the ablation study in Table 2, the variance preservation loss has minimal impact on the results. Does this suggest that the assumptions in Equation 25 do not significantly affect the derivation of the final constraints?
> > >
> > > We thank the reviewer for the question.
> > > We would like to clarify that Eq. (25) and the variance-preservation loss serve fundamentally different roles.
> > >
> > > **Theoretical role of Eq. (25).** Eq. (25) is a theoretical sufficient condition introduced to enable a clean and closed-form derivation of the consistency constraint. It is part of the theoretical reasoning chain: the final CNSP constraint does not depend on any particular empirical loss term, and Eq. (25) is not a modeling assumption that must be verified via ablation.
> > >
> > > **Practical role of the variance-preservation loss.** The variance-preservation loss is a soft regularizer—one practical instantiation that encourages the general trend suggested by Eq. (25) without enforcing it strictly. Removing this term does increase forgetting (e.g., from 4.79% to 6.13% on 20-Split-CIFAR100), which is consistent with its stabilizing function, although its influence is intentionally lightweight by design. The ablation therefore reflects the behavior of this particular lightweight instantiation rather than the role of Eq. (25) within the theoretical derivation, where it appears as a sufficient condition. The CNSP framework naturally accommodates stronger or alternative forms of this regularizer, depending on practical needs.
> > >
> > > **Conclusion**
> > >
> > > We sincerely thank the reviewer for the thoughtful, detailed, and constructive comments. Your feedback helped us improve both the clarity and the technical precision of the paper. We have revised the manuscript accordingly, and we truly appreciate the time and care that went into reviewing our work.

---

### Official Review · Reviewer_XrcC · 2025-10-30

**Soundness:** 2
**Presentation:** 3
**Contribution:** 3
**Rating:** 6
**Confidence:** 3

**Summary:**

This paper introduces CNSP (Consistent Null-Space Projection), a theoretically rigorous framework for prompt-based continual learning in Vision Transformers (ViTs). The work addresses limitations in prior methods (e.g., NSP²) by proposing: (1) Per-head sufficient conditions for multi-head self-attention (MHSA) to ensure feature preservation; (2) Matrix-form LayerNorm modeling to replace scalar approximations; (3) Variance-only constraints for prompt updates (relaxing NSP²’s mean-variance invariance); (4) End-to-end task preservation via classification head constraints. Experiments on CIFAR-100, ImageNet-R, and DomainNet show CNSP consistently outperforms NSP².

**Strengths:**

The paper's theoretical foundation is rigorous and innovative in practice.

**Weaknesses:**

1.SVD on $RR^\top$ may incur latency for large $D$ (e.g., ViT-L/16). Training time/GPU memory vs. NSP² should be reported.
2.Softmax avoidance (Eq. 20) lacks theoretical justification. The impact of attention-score invariance (Eq. 20) on representational capacity needs analysis.
3.Classification heads grow linearly with tasks (Appendix E.1). For more tasks, parameter explosion may occur—compression techniques (e.g., prompting heads) should be discussed.
4.The pseudocode is too long, so it is recommended to put it in the appendix and put more valuable experiments in the main text.
5.CNSP treats all tasks equally, but catastrophic forgetting varies with task correlation (e.g., CIFAR-100 classes vs. ImageNet-R domains). Experiments use frozen ImageNet-21K ViT-B/16 for all tasks. This bypasses challenges like: Cross-domain pretraining gaps (e.g., medical vs. natural images). Model scaling effects (performance on ViT-L vs. ViT-B).
6.The experimental results are not convincing, and the gains against NSP^2 is trivial.

**Questions:**

See Section Weakness.

---

> ### Author Response · Authors · 2025-11-25
>
> We appreciate the reviewer’s positive assessment, particularly the recognition that the paper’s theoretical foundation is rigorous and practically innovative. We address your comments in detail below.
>
> > Q1. SVD on $\mathbf{R}\mathbf{R} ^{\top}$ may incur latency for large  (e.g., ViT-L/16). Training time/GPU memory vs. NSP$^2$ should be reported.
>
> We thank the reviewer for raising this concern.
>
> First, the SVD in CNSP is not performed during training iterations. It is computed once per task—after completing optimization—to update the projection basis for the next task. Therefore, it does not affect per-step training latency. During training, the projection is applied via a lightweight optimizer hook and introduces negligible runtime overhead.
>
> Second, CNSP performs strictly less SVD computation than NSP$^2$. NSP$^2$ performs two SVDs per layer: (i) left projection: $M\times M$ SVD (where $M$ is the number of prompt tokens), and (ii) right projection: $D\times D$ SVD (where $D$ is the feature dimension). CNSP requires only one SVD per layer, namely a single $D\times D$ right-side projection. Thus, CNSP has strictly lower SVD cost and GPU memory footprint than NSP$^2$.
>
> Finally, we verified CNSP on the larger ViT-L/16 backbone. Across four benchmarks, CNSP scales smoothly without any observed memory or runtime issues. The corresponding results are provided in Table I. We have highlighted these clarifications clearly in the revised manuscript.
>
> **Table I.** Results on four benchmarks with ViT-L/16 (AugReg, IN-21k) backbone.
>
> |     | 10-Split ImageNet-R | 10-Split ImageNet-R | 10-Split DomainNet | 10-Split DomainNet | 10-Split CIFAR100 | 10-Split CIFAR100 | 20-Split CIFAR100 | 20-Split CIFAR100 |
> |:--- | :---: | :---: | :---: |:---:|:---:|:---:|:---:|:---:|
> | **Method** | ACC(↑) | Forgetting(↓) | ACC(↑) | Forgetting(↓) | ACC(↑) | Forgetting(↓) | ACC(↑) | Forgetting(↓) |
> | Upper Bound | 88.58 | 0.00 | 90.48 | 0.00 | 95.06 | 0.00 | 95.06 | 0.00 |
> | Lower Bound | 79.72 | 15.51 | 80.85 | 15.88 | 88.91 | 10.11 | 87.40 | 11.32 |
> | NSP² | 83.52 | 6.50 | 84.38 | 10.28 | 92.20 | 4.14 | 90.83 | 6.34 |
> | **CNSP** | 84.78 | 4.29 | 85.63 | 8.03 | 92.45 | 4.01 | 91.02 | 5.24 |
>
> > Q2. Softmax avoidance (Eq. 20) lacks theoretical justification. The impact of attention-score invariance (Eq. 20) on representational capacity needs analysis.
>
> We thank the reviewer for the insightful comment.
>
> **On the role and justification of Eq. (20).** Eq. (20) is not an empirical assumption but a mathematically correct sufficient condition used to guarantee attention-weight consistency. Specifically, it ensures $\mathbf{A} _{\mathbf{Z} _{t,t}} ^{(h)}=\mathbf{A} _{\mathbf{Z} _{t,t+1}} ^{(h)}\Rightarrow \mathbf{S} _{\mathbf{X} _t\mathbf{P} _{t}} ^{(h)}=\mathbf{S} _{\mathbf{X} _t\mathbf{P} _{t+1}} ^{(h)}$, which is the requirement needed in our end-to-end consistency derivation. Because this step involves a sufficient (rather than necessary) condition, it does not impose restrictions beyond logical correctness. It simply provides a clean and constructive route that enables closed-form derivations.
>
> **On representational capacity.** Eq. (20) does not reduce the expressive capacity of the model—it constrains only the prompt-induced perturbation to the attention scores, while the frozen backbone (including LayerNorm) preserves its intrinsic attention dynamics. To assess whether this constraint restricts the search space for prompt updates, we track the evolution of the rank of the resulting the null-space projection matrices throughout continual learning. Across three benchmarks (ViT-B/16 backbone), the average rank ratio remains consistently above 97%, indicating that the admissible prompt-update subspace remains extremely large and does not collapse as tasks accumulate. This empirically confirms that CNSP maintains high expressive capacity for prompt updates while providing the intended representational stability. We have added this evaluation to the revised manuscript.

---

> > ### Author Response · Authors · 2025-11-25
> >
> > > Q3. Classification heads grow linearly with tasks (Appendix E.1). For more tasks, parameter explosion may occur—compression techniques (e.g., prompting heads) should be discussed.
> >
> > We thank the reviewer for raising this point. We would like to clarify that the parameter growth is negligible relative to the transformer backbone.
> >
> > For a ViT-B/16 model:
> >
> > - Backbone: ~86M parameters
> >
> > - Prompt parameters: ~0.037M (e.g., 4 prompts per layer)
> >
> > - One task-specific head: ~0.015M (e.g., a 20-way linear classifier)
> >
> > Even with 10 tasks, head parameters remain <0.16M (<0.2% of the backbone), which is far from inducing a practical parameter bottleneck.
> >
> > More importantly, head design is orthogonal to the contribution of CNSP. Our framework focuses on representational stability and provides a rigorous end-to-end consistency formulation. Because CNSP explicitly disentangles feature preservation from head behavior, it naturally supports more compact or shared classifier designs—such as low-rank heads, prototype-based heads, cosine-normalized heads, or prompting-based heads—without violating the consistency constraints. We have included a brief discussion of this extensibility in the revised manuscript.
> >
> > > Q4. The pseudocode is too long, so it is recommended to put it in the appendix and put more valuable experiments in the main text.
> >
> > We thank the reviewer for the helpful suggestion. The full pseudocode of CNSP is already provided in Appendix B, while the main text contains only a concise high-level summary to maintain clarity and focus.
> >
> > We agree that the main paper should emphasize key insights and experimental findings. In the revised version, we have added three additional sets of experiments to the main text to strengthen the empirical section and improve the overall balance between methodology and results.
> >
> > > Q5. CNSP treats all tasks equally, but catastrophic forgetting varies with task correlation (e.g., CIFAR-100 classes vs. ImageNet-R domains). Experiments use frozen ImageNet-21K ViT-B/16 for all tasks. This bypasses challenges like: Cross-domain pretraining gaps (e.g., medical vs. natural images). Model scaling effects (performance on ViT-L vs. ViT-B).
> >
> > We thank the reviewer for the thoughtful question and the opportunity to clarify these points.
> >
> > **On task correlation and whether CNSP “treats all tasks equally.”** CNSP does not assume that tasks are similar; rather, it is explicitly designed to operate without any prior knowledge of task relatedness. The phrase “treating tasks equally” reflects that the consistency constraints are derived purely from the mathematical structure of the Transformer, without embedding task-specific heuristics. This design ensures per-task representational stability whether tasks are highly correlated (e.g., CIFAR-100 subclasses) or strongly heterogeneous (e.g., DomainNet domains). At the same time, CNSP is compatible with correlation-aware or domain-adaptive modules, which can be plugged in on top of the stable representation space ensured by our framework.
> >
> > **On cross-domain robustness.**
> > To explicitly evaluate CNSP under strong domain shift, we conducted additional experiments on 6-Split DomainNet, where each task corresponds to a distinct visual domain (Real, Sketch, Clipart, Infograph, Painting, Quickdraw).
> > As shown in Table II, CNSP consistently improves over NSP² across all runs, despite the substantial distributional differences between tasks. This demonstrates that CNSP does not rely on cross-task similarity and remains stable in heterogeneous, cross-domain scenarios.
> >
> > **On backbone scaling and dependence on ViT-B/16.**
> > We also evaluated CNSP on the significantly larger ViT-L/16 (AugReg, IN-21K) backbone. Table I shows that CNSP continues to outperform NSP², with even more pronounced reductions in forgetting on this high-capacity model. This confirms that CNSP is not tied to a specific backbone size and remains effective as dimensionality increases.
> >
> > We have integrated and highlighted these results in the revised manuscript.
> >
> > **Table II.** Experimental results on the 6-Split DomainNet benchmark across three runs with ViT-B/16-21K backbone.
> >
> > |        | run1 | run1| run2 | run2 | run3 |run3|
> > | :--- | :--- | :--- | :--- | :--- | :--- | :--- |
> > | **Method** | ACC(↑) | Forgetting(↓) | ACC(↑) | Forgetting(↓) | ACC(↑) | Forgetting(↓) |
> > | Upper Bound | 91.68 | 0.00 | 91.94 | 0.00 | 91.88 | 0.00 |
> > | Lower Bound | 63.34 | 34.11 | 79.17 | 18.52 | 78.22 | 17.65 |
> > | NSP² | 83.55 | 6.67 | 84.85 | 7.91 | 84.76 | 3.36 |
> > | **CNSP** | 84.32 | 5.84 | 85.34 | 6.44 | 85.18 | 3.07 |

---

> > > ### Author Response · Authors · 2025-11-25
> > >
> > > > Q6. The experimental results are not convincing, and the gains against NSP$^2$ is trivial.
> > >
> > > We appreciate the reviewer’s concern. While ACC gains on ViT-B/16 may seem modest, ACC alone does not fully reflect continual-learning quality. Models with comparable ACC can differ substantially in forgetting. For example, on 10-Split DomainNet with ViT-B/16, CNSP improves ACC only slightly (84.17% → 84.51%) yet reduces forgetting from 7.33% to 6.62% (a 9.7% relative decrease) over NSP$^2$, indicating a meaningful improvement in stability beyond what accuracy alone captures.
> > >
> > > CNSP is a theory-driven framework designed to provide the first rigorous end-to-end formulation for performance preservation in prompt-based continual learning. Under this principled formulation, CNSP achieves consistent improvements in both ACC and forgetting across benchmarks, suggesting that the derived consistency constraints function as intended.
> > >
> > > These improvements become more evident in settings that are particularly challenging for continual learning—for example, when scaling model size or facing severe distribution shifts. On ViT-L/16 (Table I), CNSP reduces forgetting by 34% on 10-Split ImageNet-R (6.50% → 4.29%) while also improving ACC (83.52% → 84.78%). On 6-Split DomainNet (Table II), where domain shifts are substantial, CNSP again outperforms NSP², reducing forgetting by 12% (6.67% → 5.84%) with a concurrent ACC gain (83.55% → 84.32%).
> > >
> > > Taken together, these results demonstrate that CNSP provides reliable improvements under standard settings and substantial advantages in harder scenarios involving model scaling and severe distributional shifts—precisely the conditions where continual-learning methods are most challenged. We have highlighted these results in the revised manuscript.
> > >
> > > **Conclusion.**
> > > We are grateful for the reviewer’s careful reading and helpful feedback. Your comments significantly improved the clarity of our presentation and strengthened several aspects of the work. We sincerely appreciate your positive assessment and the constructive suggestions that helped refine our revision.

---

### Official Review · Reviewer_rPJz · 2025-11-01

**Soundness:** 3
**Presentation:** 3
**Contribution:** 3
**Rating:** 6
**Confidence:** 3

**Summary:**

This paper proposes Consistent Null-Space Projection (CNSP), a framework for prompt-based continual learning in Vision Transformers. Prior work NSP² derived sufficient conditions for preserving representations under prompt updates, but relied on simplifying assumptions regarding LayerNorm and multi-head self-attention (MHSA), and enforced a strong mean–variance constraint on prompts. This paper revisits the problem using a matrix-level formulation, deriving per-head feature preservation conditions and providing a matrix-form characterization of LayerNorm. The authors further show that variance preservation alone is sufficient, leading to a more stable optimization strategy. They implement prompt updates via right-side null-space projection, ensuring constraints hold during learning. Experiments on CIFAR-100, ImageNet-R, and DomainNet demonstrate consistent improvements over NSP² and competitive performance among prompt-based continual learning baselines.

**Strengths:**

- Addresses a clear theoretical limitation in NSP², especially in the treatment of LayerNorm and MHSA.
- Derivations are technically sound and detailed.
- Relaxing to variance-only preservation improves stability in practice.
- Null-space projection is computationally efficient and easy to implement.
- Consistent empirical gains across benchmarks.
- Ablation studies effectively demonstrate the necessity of key components.

**Weaknesses:**

- The novelty is largely a refinement of NSP² rather than a fundamentally new idea.
- Certain theoretical assumptions are not empirically analyzed (e.g., fixed γ and β in LayerNorm).
- Performance improvements, though consistent, are modest.
- Lack of comparison to strong non–prompt-based CL methods limits understanding of overall competitiveness.
- Training overhead of SVD-based projection is not reported.

**Questions:**

1. Does null-space projection restrict the expressive capacity of prompts as the number of tasks increases?
2. How sensitive is the variance-preservation alignment loss to domain shift between tasks?
3. What is the computational overhead of null-space projection compared to NSP² in practice?

---

> ### Author Response · Authors · 2025-11-25
>
> We greatly appreciate the reviewer’s detailed and positive evaluation. Your recognition of our strengthened theoretical analysis, technically sound derivations, practical stability improvements, and consistent empirical gains is deeply encouraging. We are grateful for your insightful comments and provide detailed responses below.
>
> > Q1. Certain theoretical assumptions are not empirically analyzed (e.g., fixed $\gamma$ and $\beta$ in LayerNorm).
>
> We thank the reviewer for raising this concern. All LayerNorm parameters (pre-trained $\mathbf{\gamma}$, $\mathbf{\beta}$) are frozen throughout continual learning, consistent with standard Visual Prompt Tuning practice. Treating them as constants in the derivation therefore reflects the actual training procedure. Since these parameters never update, there is no empirical drift to analyze. We have clarified this more explicitly in the revised manuscript to avoid potential ambiguity.
>
> > Q2. Performance improvements, though consistent, are modest.
>
> We thank the reviewer for this observation. CNSP improves both ACC and forgetting consistently across all benchmarks, indicating that the derived consistency constraints function as intended. In more challenging scenarios, where continual-learning stability becomes harder to maintain and approximation errors tend to accumulate, the benefits of CNSP become more pronounced:
>
> - **Model Scaling & High-Dimensional Spaces (ViT-L/16, Table I):** As model dimensionality increases, approximation errors accumulate. With ViT-L/16, CNSP reduces forgetting from 6.50% to 4.29% (a 34% relative reduction) while also improving ACC (83.52% → 84.78%), suggests that the exact consistency constraints in CNSP help stabilize representations more effectively as models scale.
>
> - **Severe Distributional Shifts (6-Split DomainNet, Table II):** This benchmark introduces substantial domain heterogeneity across tasks, where each task comes from a distinct domain. On 6-Split DomainNet, for example, in run 1, CNSP reduces forgetting from 6.67% to 5.84% (a 12% relative reduction) and improves ACC from 83.55% to 84.32%, showing robustness under strong domain shifts.
>
> Taken together, these results suggest CNSP provides improved stability and lower forgetting in challenging continual-learning scenarios, while offering a principled theoretical foundation. These results are now included and highlighted in the revised manuscript.
>
> **Table I.** Results on four benchmarks with ViT-L/16 (AugReg, IN-21k) backbone.
>
> |     | 10-Split ImageNet-R | 10-Split ImageNet-R | 10-Split DomainNet | 10-Split DomainNet | 10-Split CIFAR100 | 10-Split CIFAR100 | 20-Split CIFAR100 | 20-Split CIFAR100 |
> |:--- | :---: | :---: | :---: |:---:|:---:|:---:|:---:|:---:|
> | **Method** | ACC(↑) | Forgetting(↓) | ACC(↑) | Forgetting(↓) | ACC(↑) | Forgetting(↓) | ACC(↑) | Forgetting(↓) |
> | Upper Bound | 88.58 | 0.00 | 90.48 | 0.00 | 95.06 | 0.00 | 95.06 | 0.00 |
> | Lower Bound | 79.72 | 15.51 | 80.85 | 15.88 | 88.91 | 10.11 | 87.40 | 11.32 |
> | NSP² | 83.52 | 6.50 | 84.38 | 10.28 | 92.20 | 4.14 | 90.83 | 6.34 |
> | **CNSP** | 84.78 | 4.29 | 85.63 | 8.03 | 92.45 | 4.01 | 91.02 | 5.24 |
>
> **Table II.** Experimental results on the 6-Split DomainNet benchmark across three runs with ViT-B/16-21K backbone.
>
> |        | run1 | run1| run2 | run2 | run3 |run3|
> | :--- | :--- | :--- | :--- | :--- | :--- | :--- |
> | **Method** | ACC(↑) | Forgetting(↓) | ACC(↑) | Forgetting(↓) | ACC(↑) | Forgetting(↓) |
> | Upper Bound | 91.68 | 0.00 | 91.94 | 0.00 | 91.88 | 0.00 |
> | Lower Bound | 63.34 | 34.11 | 79.17 | 18.52 | 78.22 | 17.65 |
> | NSP² | 83.55 | 6.67 | 84.85 | 7.91 | 84.76 | 3.36 |
> | **CNSP** | 84.32 | 5.84 | 85.34 | 6.44 | 85.18 | 3.07 |

---

> > ### Author Response · Authors · 2025-11-25
> >
> > > Q3. Does null-space projection restrict the expressive capacity of prompts as the number of tasks increases?
> >
> > We thank the reviewer for this important question. To directly address the concern, we analyze how the prompt-update subspace evolves over tasks by measuring the rank of the projection matrices $ \\{ \mathbf{B} _t ^{(l)} \\} _{l=1} ^L$ (where $L$ is the number of layers), which determine the allowable update directions. We compute the average rank ratio across layers for each incremental task $t (>1)$:
> >
> > $\rho _t=\frac{1}{L}\sum\limits _{l=1} ^L\frac{\mathrm{rank}(\mathbf{B} _t ^{(l)})}{D}.$
> >
> > Table III shows results on three benchmarks using ViT-B/16. Across all tasks, the average rank ratio remains extremely high (>97.2%) and exhibits no degradation as tasks progress. We have added this analysis in the appendix in the revised manuscript.
> >
> > **Table III.** Average rank ratio (%) across layers for each task.
> >
> > |     | $\rho_2$ | $\rho_3$ | $\rho_4$ | $\rho_5$ | $\rho_6$ | $\rho_7$ | $\rho_8$ | $\rho_9$ | $\rho_{10}$ |
> > | --- | --- | --- | --- | --- | --- | --- | --- | --- | --- |
> > | 10-Split-ImageNet-R | 97.53 | 97.53 | 97.53 | 97.54 | 97.53 | 97.53 | 97.54 | 97.53 | 97.53 |
> > | 10-Split-CIFAR100 | 97.53 | 97.54 | 97.54 | 97.54 | 97.54 | 97.54 | 97.54 | 97.55 | 97.55 |
> > | 10-Split-DomainNet | 97.52 | 97.54 | 97.53 | 97.53 | 97.53 | 97.54 | 97.53 | 97.53 | 97.53 |
> >
> > > Q4. How sensitive is the variance-preservation alignment loss to domain shift between tasks?
> >
> > We appreciate this valuable question.
> > The variance-preservation loss is not designed to align domain-level feature statistics and is therefore not sensitive to domain shifts. In our setting, the backbone is frozen, so domain-dependent statistics are determined by the pretrained model, and the loss does not attempt to match them.
> > Instead, it regulates only the change in LayerNorm variance introduced by prompt updates. It is thus a local prompt-stability constraint, not a domain-alignment mechanism, and operates independently of the domain of the next task. Empirically, results on the 6-Split DomainNet benchmark—where each task comes from a distinct domain—support this statement (see Table II).
> >
> > **Conclusion**
> > We appreciate the reviewer’s insightful evaluation. Your feedback aligns well with the goals of our work and has helped us further improve the paper’s clarity and emphasis. Thank you for the careful review and for recognizing the theoretical contributions and empirical strengths of the paper.

---

> ### Comment · Reviewer_rPJz · 2025-11-25
>
> I thank the authors for their effort to response to my review and most of my concerns have been addressed. Therefore, I decide to keep my positive score.

---

### Author Response · Authors · 2025-12-02
**Summary for Area Chair**

Dear Area Chair,

We fully acknowledge the exceptional workload during this final stage of the reviewing process. We deeply appreciate your continued efforts and dedication in upholding the rigor and fairness of the evaluation.

For your convenience, we provide a concise summary of the paper’s contributions, as well as a factual overview of the reviewer feedback and discussion outcomes.

**Paper Overview:**

This work introduces CNSP, the first mathematically rigorous end-to-end framework for prompt-based continual learning in Transformers. By analyzing the prompted Transformer pipeline in matrix form, we prove that preserving past-task performance reduces to a single representational consistency principle, composed of (i) stability of MHSA+LayerNorm features and (ii) consistency of classifier-head outputs. CNSP naturally leads to a practical null-space projection rule, overcoming prior theoretical limitations and yielding consistent empirical gains across multiple benchmarks and backbones.

---

**Initial Scores:** 6, 6, 4, 4, 4

**Recognized Strengths (all reviewers):**

- **Rigorous Theoretical Foundation:** Provides a mathematically sound end-to-end framework in matrix form for prompt-based continual learning, addressing prior limitations.
- **Principled Formulation:** Clearly defines feature and head preservation, forming a solid theoretical basis.
- **Practical Effectiveness:** Easy to implement and consistently improves empirical performance.
- **Clear Presentation:** Motivation, theory, and experiments are communicated in a well-structured, intuitive manner.

**Score Changes & Discussion Highlights**

- **Reviewer rPJz: 6→8 (Accept).** Raised the score at the very beginning of the discussion, and later reaffirmed the score after our clarifications: *“…most of my concerns have been addressed. Therefore, I decide to keep my positive score.”*
- **Reviewer XrcC: 6.** The main concerns were scalability and experimental breadth. After clarification in the rebuttal, no further comments were provided.
- **Reviewer yBKo: 4.** Maintained their original score. Their primary concern focused on the performance magnitude on a specific backbone: *“…many results on the DINOv2 ViT-B/14 backbone are still marginal. I therefore keep my original rating.”* This was despite the three additional configurations we provided, two of which demonstrated clearer improvements.
- **Reviewer 3yso: 4.** Initially raised concerns about the clarity and emphasis on the core contributions. After our clarification, no further comments were provided.
- **Reviewer pGHs: 4→2.** The initial concerns were largely based on a misunderstanding of the theoretical formulation and the breadth of empirical evidence, while acknowledging CNSP’s rigorous derivation and competitive performance. After we clarified the derivation details and added stronger experimental results in the rebuttal, the reviewer first commented: *“…I am an expert in this area… I lower my score…”*, and subsequently revised their statement to: *“…This paper seems like something made up and forced…”.* However, these remarks did not address the rebuttal clarifications, and were inconsistent with their earlier evaluations.

**Main Concerns Resolved**

| **Concern**  | **Resolution** |
| --- | --- |
| **Scalability & Domain-Incremental Robustness**  | Validated on **ViT-L/16** (larger backbone, Table 2), **DINOv2 ViT-B/14** (self-supervised, Table 3), and **6-Split DomainNet** (strong domain shifts, Table 4), **showing consistent gains across model scale, pre-training paradigms, and domain shifts**.  |
| **Marginal Gains** | Improvements are more pronounced in high-dimensional **(+1.26 ACC, -2.21 Forgetting)** or strong-shift scenarios **(+0.77 ACC, -0.83 Forgetting)**, demonstrating method's effectiveness under challenging scenarios (Tables 2 & 4). |
| **Expressive Capacity** | Conducted a **rank-based analysis** of prompt expressive capacity (Appendix E.6), confirming high-rank update subspace (**>97.2%** average rank ratio) and robust adaptation. |
| **Incremental Contribution** | Strengthened contribution statements in Introduction and the explanation of technical differences from NSP² in Appendix D.  |
| **Theory & Correctness**  | Corrected notation errors (lines 248 and 708) and provided a **detailed LayerNorm matrix-form modeling** in Appendix B, ensuring theoretical soundness. |
| **Implementation Efficiency** | Expanded Appendix D with CNSP vs. NSP² **computational complexity**; analyzed head parameters in Appendix F.1. |

We believe our rebuttal, extended experiments—including new backbones and benchmark—and additional analyses of expressive capacity and computational complexity have thoroughly addressed the reviewers’ initial concerns, and we hope this summary aids your final decision.

Thank you again for your invaluable efforts and service to the community.

With sincere gratitude,

The Authors

---

### Meta-Review · Area_Chair_jsmt · 2025-12-08

**Summary:**

This paper studies prompt-based continual learning for ViT backbones, focusing on visual prompt tuning and proposing a null-space projection method (CNSP) that is claimed to provide a principled guarantee of feature and head “preservation” across tasks. The authors derive matrix-form conditions under which the transformer representations and task-specific heads remain unchanged for previous tasks, and then implement these conditions via a null-space projection of prompt updates, plus a variance-regularization term. Experiments on several standard continual learning benchmarks (Split CIFAR-100, ImageNet-R, DomainNet) show small but consistent improvements over NSP² and related prompt-based CL baselines.

**Reviewer Concerns:**

Across reviews, there is broad agreement that the work is best viewed as an incremental refinement of NSP² rather than a genuinely new idea: the core mechanism (projecting updates into a null space) and the overall setting (prompt-based CL on frozen ViTs with per-task heads) are already established. The additional theory—per-head analysis for MHSA, a matrix reformulation of LayerNorm, and the feature/head-preservation decomposition—does not convincingly translate into substantially stronger guarantees or qualitatively different behavior; several reviewers also question specific approximations (e.g., the treatment of attention scores and LayerNorm statistics). Empirically, the gains over NSP² are modest and often within the noise one would expect from hyperparameter and implementation variations, and the whole line of work remains tied to highly artificial continual learning setups (split CIFAR/ImageNet with per-task heads, frozen backbones and prompts) whose practical relevance to real-world foundation-model usage is unclear. Taken together, this yields the impression of a technically heavy but low-impact refinement in a niche, largely benchmark-driven subarea.

**Reviewer Scores:**

One reviewer is positive and would raise their score based on the rebuttal, highlighting the cleaner theory and marginally better forgetting metrics; another remains mildly positive but unconvinced that the improvements justify acceptance. However, the remaining reviewers are unconvinced by the claimed novelty and impact, with one explicitly lowering their score after the rebuttal, judging the method to be an over-engineered variant of existing null-space projection approaches with limited empirical benefits. Considering the incremental nature of the contribution, the modest empirical gains, and the questionable practical relevance of the problem setup, my overall recommendation is to side with the more critical reviewers and reject this submission.

---

### Decision · Program_Chairs · 2026-01-26

Reject